# Reconstructing the EFT of inflation from cosmological data

**Amel Durakovic[1,2]⋆, Paul Hunt[†], Subodh P. Patil[1][‡] and Subir Sarkar[3]∘**

**1** Niels Bohr International Academy & Discovery Center, Niels Bohr Institute,
Blegdamsvej 17, Copenhagen, DK 2100, Denmark
**2** Technical University of Denmark, Department of Physics, Fysikvej,
Building 309, Kongens Lyngby, DK 2800, Denmark
**3** Rudolf Peierls Centre for Theoretical Physics, University of Oxford,
Parks Road, Oxford OX1 3PU, United Kingdom

⋆ ameld@dtu.dk, † paul.hunt@fuw.edu.pl, ‡ patil@nbi.ku.dk, ∘ s.sarkar@physics.ox.ac.uk

## Abstract

Reconstructions of the primordial power spectrum (PPS) of curvature perturbations from cosmic microwave background anisotropies and large-scale structure data suggest that the usually assumed power-law PPS has localised features (up to $\sim 10\%$ in amplitude), although of only marginal significance in the framework of $\Lambda$CDM cosmology. On the other hand if the cosmology is taken to be Einstein-de Sitter, larger features in the PPS (up to $\sim 20\%$ in amplitude) are required to accurately fit the observed acoustic peaks. Within the context of single clock inflation, we show that any given reconstruction of the PPS can be mapped on to functional parameters of the underlying effective theory of the adiabatic mode within a 2nd-order formalism, provided the best fit fractional change of the PPS, $\Delta\mathcal{P}_{\mathcal{R}}/\mathcal{P}_{\mathcal{R}}$ is such that $(\Delta\mathcal{P}_{\mathcal{R}}/\mathcal{P}_{\mathcal{R}})^3$ falls within the $1\sigma$ confidence interval of the reconstruction for features induced by variations of either the sound speed $c_{\rm s}$ or the slow-roll parameter $\epsilon$. Although there is a degeneracy amongst these functional parameters (and the models that project onto them), we can identify simple representative inflationary models that yield such features in the PPS. Thus we provide a dictionary (more accurately, a thesaurus) to go from observational data, via the reconstructed PPS, to models that reproduce them to per cent level precision.

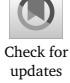
## Contents



# 1   Introduction

All observations of the cosmic microwave background (CMB) are consistent with scale invariant, adiabatic, Gaussian initial conditions [1] – widely accepted as evidence of an epoch of early universe inflation, processed by an intervening $\Lambda$CDM cosmology [2]. This is not to say that we have directly observed scale invariant, adiabatic and Gaussian initial conditions, since the most we can do with the limited set of modes we observe (already compressed by projection onto the 2-dimensional surface of last scattering) is to marginalise over or fix all but a set number of parameters, and look for the best fit for the remaining parameters consistent with the observed CMB sky, as with the six parameter $\Lambda$CDM model. It is then necessary to introduce theoretical priors motivated by an underlying model bias for this process to work. A common such set of priors involves parameterising the spectrum of curvature perturbations as a power-law, thus modelling it with only two numbers – an amplitude at some pivot scale and a spectral index. This is a natural parameterisation in the context of toy models of single scalar field inflation. One can also allow for mild departures from this model such as a 'running' of the spectral index, or even a running of the running. However such parameterisations would not be up to task if the data has pronounced localised departures from scale-invariance i.e. there are sizeable features superimposed on the power-law spectrum. [1]

Direct reconstruction of the PPS holding all other parameters of the $\Lambda$CDM model fixed using CMB temperature data, temperature plus polarisation data, or CMB cross correlated with large scale structure [5]–[33] suggests that small, localised departures from scale invariance may indeed be present in the data, although with marginal statistical significance. On the other hand, attempts to match CMB observations with an Einstein-de Sitter (EdS) cosmology (which has $\Omega_\Lambda = 0$) requires a PPS with larger departures from scale invariance in order to reproduce the observed acoustic peaks in the CMB power spectrum [36].

This begs the question – what underlying model of inflation could have produced the requisite features? Those needed when an underlying EdS cosmology is assumed can be naturally generated in 'multiple inflation' in $N = 1$ supergravity wherein the inflaton mass changes

---

[1]Note that this has been invoked (at scales beyond those accessible in the CMB) as a mechanism to produce primordial black holes (see [3, 4] for reviews).

suddenly due to its (gravitational) coupling to other 'flat direction' scalar fields which undergo symmetry-breaking phase transitions *during* inflation [34,35]. The bump-like feature required in the PPS to fit an EdS universe would require two phase transitions in rapid succession — one which raises the inflaton mass and a second soon after which lowers it [36,37]. Although additional parameters then need to be introduced to describe this, it is justified by the resultant overall improvement of the fit to the data e.g. the $\chi^2$/d.o.f. according to standard information criteria. Moreover there are other observable signatures and tests of such multiple inflation, e.g. associated characteristic non-gaussianity [38]. However there may well be other ways to generate the required PPS in very different models of inflation, or more generally if one were to assume different underlying cosmologies. It would therefore be useful to provide an inflation model-*independent* description of what is required to fit the observational data.

In this investigation, we provide the tools to address this question in an arbitrary context by detailing a procedure by which one can 'invert' any given reconstructed power spectrum for a given universality class of inflationary backgrounds if the reconstructed features are small enough. By this, we mean that if the reconstructed spectrum is such that $(\Delta\mathcal{P}_\mathcal{R}/\mathcal{P}_\mathcal{R})^3 \lesssim \Sigma(k)$ everywhere – where $\Sigma(k)$ is the $1\,\sigma$ confidence interval of the reconstruction of the fractional PPS at any given scale – then a 2nd-order formalism suffices to invert for a representative background model if the features were to be induced by a variation in the adiabatic sound speed $c_\mathrm{s}$ or $\epsilon$, the first term in the Hubble hierarchy. That is, *it is possible to convert any given scale dependence for the primordial power spectrum into a time dependence for the parameters of the underlying effective theory (EFT) of the adiabatic mode* [39] in the context of single clock inflation. Although many background models will project onto the same set of parameters of the EFT of the adiabatic mode, one can always look for the simplest model that will reproduce such time dependence and nominate it as the most plausible representative of its class to reproduce such features. In other words, we present a dictionary, or rather a thesaurus, to map any given reconstructed primordial power spectrum to a class of models that would reproduce it.

As elaborated upon in the next section, all models of single clock inflation project onto at most three independent functions of time in the effective theory of the adiabatic mode up to quadratic order. Of these, $\epsilon$ and $c_\mathrm{s}$ capture the effects of the leading order terms in the derivative expansion in the parent theory. For features induced by variations in $\epsilon$ alone, one can explicitly reconstruct a potential that would reproduce the required variations in $\epsilon$ presuming a canonical kinetic term, as shown in Appendix B. We find that even $\sim 20\%$ amplitude features can be reproduced with potentials that naïvely do not appear to differ too significantly from a polynomial potential over the field range responsible for the observed modes (see Figs. 7 and 9). However, the coefficients of the potential are indeed finely tuned so that the background trajectory is intermittently knocked off the attractor, thus rendering sizeable features at the desired scale. We speculate on the microphysical origin of such potentials and their radiative stability in our concluding discussion. Although we do not consider it in detail in the present study, we mention in the following section how one can also in principle construct representative background models for reconstructed features produced by variations in $c_\mathrm{s}$ within the effectively single clock context. [2]

The outline of this paper is as follows – in § II, we review the effective theory of the adiabatic mode, detailing the method by which one can reproduce the scale dependence of a given reconstructed $\Delta\mathcal{P}_\mathcal{R}/\mathcal{P}_\mathcal{R}$ by a time dependence in $c_\mathrm{s}$ keeping $\epsilon$ fixed, or vice-versa, within a 2nd-order formalism. In § II. A we demonstrate the utility of our formalism with an analytic toy example of a $\mathcal{O}(10\,\%)$ feature and reconstruct a model potential that reproduces the feature

---

[2]Features can also be generated in the *non*-single clock context (see e.g. [40–42]), however the formalism developed here to 'invert' for representative background models does not apply to such scenarios due to added degeneracies (see also [43] for a recent discussion).

to percent level accuracy. In § III and IV, we detail a direct reconstruction of the PPS from Planck data and present results assuming first the standard ΛCDM (§ V) and then the very different EdS cosmology (§VI), again reconstructing possible background model potentials that reproduce the PPS. Finally in § VII we offer our conclusions. Various details not covered in the main body of the paper are elaborated on in the appendices.

## 2 From features to 'Wilson functions'

The underlying philosophy of effective field theory is no different to that of the Taylor expansion, where in place of expanding a function around a given point with a complete basis of functions (say monomials in the case of a single variable), one expands an effective action in terms of a complete basis of operators consistent with the symmetries of the system. The standard expansion for a Lorentz covariant theory takes the form of a derivative expansion in canonically normalised fields [44, 45], whose coefficients are to be fixed by a finite number of measurements at some fixed energy scale. Operators with mass dimension greater than four are classified as *irrelevant*, meaning that their effects at energies much lower than the mass scale defining the operator expansion are subleading. [3] The craft of effective field theory consists of choosing a suitable operator basis consistent with the symmetries of the system such that quantum corrections do not generate large anomalous dimensions to these operators. That is, the bare Lagrangian one writes down describes the propagation of degrees of freedom that reasonably approximate the true quantum mechanical degrees of freedom. In the context of adiabatic cosmology, the situation is complicated by the fact that the background spontaneously breaks Lorentz invariance (increasing the number of operators one can write down consistent with the remaining symmetries). A suitable operator basis that has come to be known as 'the Effective Theory of Inflation' was proposed in [39] (see also [46] for a review with an eye to some of the applications presented here).

The insight of [39] was to exploit the redundancy inherent in a diffeomorphism invariant theory to foliate spacetime in such a way that the fluctuations of the scalar degree of freedom that generated the cosmological background are gauged away. In this so-called comoving (or unitary) gauge, the fluctuating background source has now been absorbed into the metric, which acquires a propagating scalar polarisation – the comoving curvature perturbation $\mathcal{R}$, defined via the 3-metric

$$h_{ij} = a^2 e^{2\mathcal{R}} \delta_{ij}. \tag{1}$$

Together with the lapse and shift vectors specifying the foliation ($N$ and $N^i$ respectively) this completely characterises the metric via the ADM decomposition

$$ds^2 = -N^2 dt^2 + h_{ij}\left(dx^i + N^i dt\right)\left(dx^j + N^j dt\right). \tag{2}$$

The comoving curvature perturbation is the workhorse of the effective theory – it is an ungapped Goldstone mode that non-linearly realises time translation invariance in single clock cosmology. This has two important implications. Firstly, because it is ungapped, $\mathcal{R}$ can only have derivative couplings, meaning that at long wavelengths $\mathcal{R} \equiv$ constant will always be a solution to any order in perturbation theory [47], a statement that can also be proved at the quantum level [48]. This is the familiar constant super-horizon mode that imprints on the CMB around last scattering. The second important feature is that the coefficients of any operators one writes down at different orders have non-trivial relations forced upon them by the

---

[3]This mass scale is often referred to as the cutoff of the effective theory, but is better thought of as the scale at which new physics become relevant, necessitating another effective description that possibly includes propagating heavier degrees of freedom not included in the original description.

non-linear realisation. This means that the 'Wilson functions' that determine the expansion at quadratic order necessarily imprint on higher order correlation functions as well. This has the corollary that any features in the two-point function of the curvature perturbation will correlate with features in the three-point function in a manner that can be quantified precisely if the feature is due to variations in $c_s$ [49], or more generally [50, 51], with additional consistency relations forced upon higher-point correlation functions [52, 53].

The operator basis defining the EFT of inflation is given by

$$S = \int d^4x \sqrt{-g}\left[ M_{\rm pl}^2 \frac{R^{(4)}}{2} - M_{\rm pl}^2\left(\frac{\dot{H}}{N^2} - 3H^2 - \dot{H}\right) + \frac{M_2^4}{2!}(\delta g^{00})^2 + \frac{M_3^4}{3!}(\delta g^{00})^3 + \ldots \right.$$
$$\left. + \widehat{M}_2^3 \delta g^{00} \delta E_i^i + \frac{\widetilde{M}_2^2}{2!}\left(\delta E_i^i\right)^2 + \frac{\bar{M}_2^2}{2!}\left(\delta E^{ij}\delta E_{ij}\right)^2 + \ldots \right], \tag{3}$$

where $\delta g^{00} = g^{00} + 1$ and $\delta E_{ij}$ is the variation of $E_{ij}$ which is related to the extrinsic curvature $K_{ij}$ of the hypersurfaces defining the foliation as

$$E_{ij} = NK_{ij} = \frac{1}{2}\left(\dot{h}_{ij} - \nabla_i N_j - \nabla_j N_i\right), \tag{4}$$

with the ellipses denoting higher order terms. [4] One obtains the action up to the $n^{\rm th}$ power of $\mathcal{R}$ by solving for the lapse and shift constraints to the $(n-2)^{\rm th}$ power [54, 55] and substituting back into the action. Only the following four operators in the EFT expansion can contribute terms quadratic in $\mathcal{R}$:

$$\mathcal{L}_{(2)} \sim (\delta g^{00})^2,\ \delta g^{00}\delta E_i^i,\ (\delta E_i^i)^2,\ \delta E^{ij}\delta E_{ij}, \tag{5}$$

where the last two operators give equivalent contributions at quadratic order after integration by parts. Therefore, considering only the operators $(\delta g^{00})^2,\ \delta g^{00}\delta E_i^i,\ (\delta E_i^i)^2$ with coefficient (Wilson) functions $M_2^4(t), \widehat{M}_2^3(t)$ and $\widetilde{M}_2^2(t)$ respectively, one obtains the following 2nd-order action after solving for the lapse and shift constraints:

$$S_2 = \int d^4x\ a^3 \epsilon M_{\rm pl}^2 \left(\frac{\dot{\mathcal{R}}^2}{c_s^2} - \frac{(\partial\mathcal{R})^2}{a^2} + \mu^{-2}\frac{(\partial^2\mathcal{R})^2}{a^4}\right), \tag{6}$$

where in general $c_s$ and $\mu(t)$ are complicated functions of the three Wilson functions $M_2^4(t), \widehat{M}_2^3(t)$ and $\widetilde{M}_2^2(t)$. It can be shown that the functional coefficient of the operator $(\partial^2\mathcal{R})^2$ can only be generated by the $(\delta E_i^i)^2$ term [68]. However, certain simplifications occur if the parent matter effective action describing the inflaton background takes the form

$$\mathcal{L}_m = \mathcal{L}_m(\phi, \partial\phi), \tag{7}$$

i.e. contains only derivatives that come in combinations of the form $(\partial\phi)^{2n}$ in its effective expansion. In this case, it is straightforward to see that operators of the form $(\delta E_i^i)^2$ or $\delta g^{00}\delta E_i^i$ will not be generated, since factors of the shift vector cannot appear in powers of $(\partial\phi)^{2n}$ in the comoving gauge, from which it directly follows that $\mu^{-2}(t) \equiv 0$. [5] However even if such

---

[4]The terms in the round parentheses enforce tapdole cancellation – i.e. guarantee that the background one expands around satisfies the Friedmann equations. If we were to calculate loop corrections to this action, the tadpole condition would shift accordingly i.e. the background equations of motion are an extremum of the effective action.

[5]This will no longer be true if operators containing factors of $\Box\phi$ appear in the parent matter effective action – e.g. at the six derivative level, if operators of the form $(\partial\phi)^2\Box^2\phi$ appear in addition to $(\partial\phi)^6$. However, it is straightforward to show [57] that up to four derivative terms, one can always bring the action of the parent matter theory into the form (7) even in the presence of couplings to much heavier fields.

operators were present, from the perspective of the parent theory $\mu^{-2}$ corresponds to a mass scale associated with higher dimensional operators, and will be sub-leading for sufficiently low energies.

Therefore, considering only the operator $(\delta g^{00})^2$ with Wilson coefficient $M_2^4(t)$, which captures the leading order behavior of higher dimensional operators in the parent theory, one finds

$$S_2 = M_{\text{pl}}^2 \int d^4x \, a^3 \epsilon \left( \frac{\dot{\mathcal{R}}^2}{c_s^2} - \frac{(\partial \mathcal{R})^2}{a^2} \right), \tag{8}$$

with

$$\epsilon = -\frac{\dot{H}}{H^2}, \quad \frac{1}{c_s^2} = 1 - \frac{2M_2^4}{M_{\text{pl}}^2 \dot{H}}. \tag{9}$$

That is, to leading order in the derivative expansion, one finds that the functions $\epsilon$ and $c_s$ paramaterise all possible single clock backgrounds (slow roll or not, canonical or not), with $\mu$ capturing subleading effects from higher dimensional operators in the parent theory. Of these functions, $\epsilon$ plays a privileged role. It is akin to an order parameter that book-keeps the expansion – when it vanishes, a symmetry is restored (exact time translational invariance) and each term in the perturbative expansion for $\mathcal{R}$ is suppressed by sequentially higher orders in $\epsilon$ [54]. Any modifications to the zero and two derivative parts of the parent matter effective action will manifest in changes in $\epsilon$. The interpretation of $c_s$ (equivalently, $M_2^4$) from the perspective of the background theory is that it captures the leading order modifications to the two and four derivative terms in the parent matter effective action.

Although the whole point of effective field theory is to be agnostic about the underlying high energy description, it is useful to illustrate the significance of the coefficient $M_2^4$ by computing it in a specific setup. For example, in effectively single field inflation, where the inflaton is a single light direction in a multi-field space where all other directions are much heavier, [6] one finds that:

$$\frac{1}{c_s^2} = 1 + \frac{4\dot{\theta}^2}{M_{\text{eff}}^2}. \tag{10}$$

Here $\dot{\theta} = V_N / \dot{\phi}_0$ is the angular velocity in field space given background field velocity $\dot{\phi}_0 = (\dot{\phi}_a \dot{\phi}^a)^{1/2}$, with $M_{\text{eff}}^2 = V_{NN} - \dot{\theta}^2$, and $V_N = N^a \nabla_a V$ is the derivative of the potential normal to the trajectory which vanishes when the inflaton is on the potential trough [58]. Intuitively, analogous to a bob-sledder going down a track, the background trajectory slides up the valley of the potential each time it traverses a bend in field space, resulting in transient reductions in the speed of sound, thus capturing the leading order effects of higher dimensional operators in the parent theory [59,60]. This can occur without spoiling slow roll [61,62] and is consistent with the decoupling of the true fast and slow modes of the theory (which no longer align with the tangent and the normal to the trough of the background potential [57]). Intuitively, effectively single field inflation corresponds to 'sliding up' the heavy directions without exciting normal oscillations (in contrast to models referred to in [40–42], where a heavier clock field no longer decouples). [7]

Comparing (9) and (10), we thus read off:

$$M_2^4 = \frac{\dot{\phi}_0^2 \dot{\theta}^2}{M_{\text{eff}}^2}. \tag{11}$$

---

[6]Specifically, in a two field setting, if $T^a$ and $N^a$ are the tangent and normal vectors to the background trajectory $\phi_0^a(t)$, then one is in the effectively single field regime when $T^a T^b \nabla_a \nabla_b V \ll N^a N^b \nabla_a \nabla_b V$.

[7]Although we do not pursue it further here, one can envisage reconstructing a potential over any given target space where the background trajectory turns in such a way that it reproduces the reconstructed $c_s$ (13).

As reviewed in Appendix A, any changes in the speed of sound sourced by a time varying $M_2^4$ can be shown to induce a change in the power spectrum to 1st-order of the form

$$\frac{\Delta_1 \mathcal{P}_\mathcal{R}}{\mathcal{P}_\mathcal{R}}(k) = -k \int_{-\infty}^{0} d\tau \left(1 - \frac{1}{c_s^2}\right) \sin(2k\tau) , \qquad (12)$$

where $\mathcal{P}_\mathcal{R}$ denotes the power spectrum of the fiducial attractor of which we consider the feature a perturbation, and where we for now work to 1st order in the quantity $u(\tau) \equiv 1/c_s^2 - 1$, hence the subscript. As shown in [49] and rederived in Appendix A, any features imprinted by transient reductions in the speed of sound to 1st-order can in principle be 'inverted' so that one could also reconstruct the function $M_2^4(t)$ that would have generated such features if they were sourced by transient reductions in the speed of sound alone

$$\frac{1}{c_s^2} - 1 = \frac{1}{\pi} \int_{-\infty}^{0} \frac{dk}{k} \frac{\Delta_1 \mathcal{P}_\mathcal{R}}{\mathcal{P}_\mathcal{R}}(k) \sin(-2k\tau). \qquad (13)$$

This suggests that provided the feature is small enough, one can always map any non-trivial scale dependence of the primordial power spectrum onto the time dependence of the parameters of the EFT of the adiabatic mode. This of course is not a unique prescription since there are many ways one can produce the same scale dependence of the two point function of the curvature perturbation given the independent functions $\epsilon$ and $c_s$. A similar exercise keeping $c_s$ fixed at unity also allows us to calculate the change in the power spectrum induced by a varying $\epsilon$. To 1st-order in $\Delta\epsilon/\epsilon$ one can show that

$$\frac{\Delta_1 \mathcal{P}_\mathcal{R}}{\mathcal{P}_\mathcal{R}}(k) = \frac{1}{k} \int_{-\infty}^{0} \frac{d\tau}{\tau^2} \frac{\Delta\epsilon}{\epsilon}(\tau)((1 - 2k^2\tau^2)\sin(2k\tau) - 2k\tau\cos(2k\tau)) . \qquad (14)$$

With a bit more work, we can also invert the integral kernel above to find the time dependence of $\Delta\epsilon/\epsilon$ corresponding to any given $\Delta_1 \mathcal{P}_\mathcal{R}/\mathcal{P}_\mathcal{R}$ (45):

$$\frac{\Delta\epsilon}{\epsilon}(\tau) = \frac{2}{\pi} \int_{0}^{\infty} \frac{dk}{k} \frac{\Delta_1 \mathcal{P}_\mathcal{R}}{\mathcal{P}_\mathcal{R}}(k) \left(\frac{2\sin^2(k\tau)}{k\tau} - \sin(2k\tau)\right). \qquad (15)$$

It turns out that some remarkable simplifications enable us to extend this inversion to 2nd-order. The fractional change in the power spectrum to 2nd-order in the EFT parameters $\Delta\epsilon/\epsilon(\tau)$ or $u(\tau) = 1/c_s^2(\tau) - 1$ (henceforth denoted $X(\tau)$ in general) is

$$\frac{\Delta \mathcal{P}_\mathcal{R}}{\mathcal{P}_\mathcal{R}}(k) = \frac{\Delta_1 \mathcal{P}_\mathcal{R}}{\mathcal{P}_\mathcal{R}}(k) + \frac{\Delta_2 \mathcal{P}_\mathcal{R}}{\mathcal{P}_\mathcal{R}}(k) , \qquad (16)$$

where the 2nd-order term has the form

$$\frac{\Delta_2 \mathcal{P}_\mathcal{R}}{\mathcal{P}_\mathcal{R}}(k) = \int_{-\infty}^{0} d\tau_2 X(\tau_2) \int_{-\infty}^{\tau_2} d\tau_1 X(\tau_1) \mathcal{K}(k, \tau_1, \tau_2) . \qquad (17)$$

The full expression for the integral kernel $\mathcal{K}$ can be found in [80]. Given a reconstruction estimate $\Delta \mathcal{P}_{\text{rec}}/\mathcal{P}_\mathcal{R}$ for the fractional change in the power spectrum, we wish to invert (16) for the EFT parameters $\Delta\epsilon/\epsilon(\tau)$ or $u(\tau)$.

Fortunately, as shown in Appendix A, the 2nd-order order fractional change in the power spectrum for features induced by varying $c_s$ (46) or $\epsilon$ is in fact equal to the square of the 1st-order fractional change for both cases (cf. (47) and (50))

$$\frac{\Delta_2 \mathcal{P}_\mathcal{R}}{\mathcal{P}_\mathcal{R}}(k) = \left(\frac{\Delta_1 \mathcal{P}_\mathcal{R}}{\mathcal{P}_\mathcal{R}}(k)\right)^2 , \qquad (18)$$

where we note that the right hand side could *a priori* have consisted of additional terms involving logarithmic derivatives of $\Delta_1 \mathcal{P}_\mathcal{R}/\mathcal{P}_\mathcal{R}$. Although these do not appear at 2nd-order, we do not preclude their appearance for higher order corrections.

This means that (16) is a quadratic equation in the 1st-order fractional change and can be inverted for an effective first order fractional change from a given reconstructed power spectrum as

$$\frac{\Delta_1 \mathcal{P}_\mathcal{R}}{\mathcal{P}_\mathcal{R}}(k) = \frac{1}{2}\left(-1 + \sqrt{1 + 4\frac{\Delta \mathcal{P}_{\text{rec}}}{\mathcal{P}_\mathcal{R}}(k)}\right). \tag{19}$$

Inserting this fractional change into the integrands of (13) or (15) allows us to obtain the functional parameters of the EFT of the adiabatic mode that would reproduce the reconstructed feature accurate up to the neglected terms, which we now quantify.

We note first that the minimum accuracy to which one is obliged to calculate a given quantity is set by the error with which it is determined from observations. In the context of a reconstructed power spectrum determined to within a given $1\sigma$ confidence interval, provided the higher order corrections induced by a varying parameter in the effective theory is everywhere smaller than or of the same order as the $1\sigma$ error, the 2nd-order treatment detailed above suffices. As discussed in the Appendix (53), if $\Sigma(k)$ denotes the $1\sigma$ confidence interval surrounding the fractional part of the best-fit reconstructed power spectrum, then if the neglected corrections are such that

$$\left|\frac{\Delta_3 \mathcal{P}_\mathcal{R}}{\mathcal{P}_\mathcal{R}}(k)\right|_{c_s,\epsilon} \sim \left|\frac{\Delta_1 \mathcal{P}_\mathcal{R}}{\mathcal{P}_\mathcal{R}}(k)\right|^3_{c_s,\epsilon} \lesssim \Sigma(k), \tag{20}$$

then the 2nd-order formalism is sufficiently accurate in accounting for features with a varying $c_s$ or $\epsilon$. We recall that the expression for the cubic correction is shorthand for a series of terms that could also include logarithmic derivatives of $\Delta_1 \mathcal{P}_\mathcal{R}/\mathcal{P}_\mathcal{R}$ acting on some factors (which will typically be of the same order as $\Delta_1 \mathcal{P}_\mathcal{R}/\mathcal{P}_\mathcal{R}$ itself). We note from (19) that if in addition, the reconstructed feature is such that it dips below $\Delta_1 \mathcal{P}_\mathcal{R}/\mathcal{P}_\mathcal{R} \leq -0.25$, then one is obliged to work to cubic order in perturbations in order to extract a real root for (19).[8] We conclude that we can therefore reproduce features as large as $\sim 25\%$ with less than $\sim 2\%$ error. Before turning to the specifics of reconstructing the primordial power spectrum from CMB data given different model assumptions, we illustrate the utility and accuracy of our formalism with an analytic toy example.

## 2.1 Analytic toy model

Consider the toy feature model induced by the fraction change in $\epsilon$ given by

$$\frac{\Delta \epsilon}{\epsilon}(N) = c_1 \exp\left(-\frac{(N-N_0)^2}{\sigma_1^2}\right) + c_2 (N-N_0)\exp\left(-\frac{(N-N_0)^2}{\sigma_2^2}\right), \tag{21}$$

with $c_1, c_2$ constants, which we plot in Fig. 1. Assuming (21) as the background (the red line of Fig. 2) we can compare the induced power spectrum by numerically integrating the mode equation

$$\frac{\mathrm{d}^2 \mathcal{R}_k}{\mathrm{d}N^2} + \left[3 - \epsilon(N) + \frac{\epsilon'(N)}{\epsilon(N)}\right]\frac{\mathrm{d}\mathcal{R}_k}{\mathrm{d}N} + \left(\frac{k}{aH}\right)^2 \mathcal{R}_k = 0, \tag{22}$$

with the results obtained from the analytic expressions for the fractional change of the power spectrum (given by (14) and (18)) under a particular time varying $\epsilon$. The 1st-order correction

---

[8]Alternatively, the attractor PPS can be lowered to reduce the deficit so that the fractional change does not go below $-0.25$.

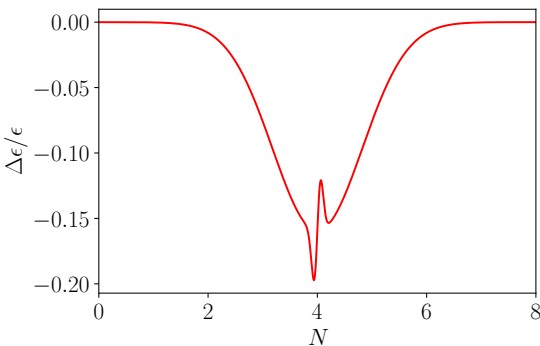

Figure 1: Red line: fractional change $\Delta\epsilon/\epsilon$ (21) with parameters $c_1 = -0.159$, $c_2 = 0.99$, $\sigma_1 = 1.16$, $\sigma_2 = 0.09$ and $N_0 = 4$.

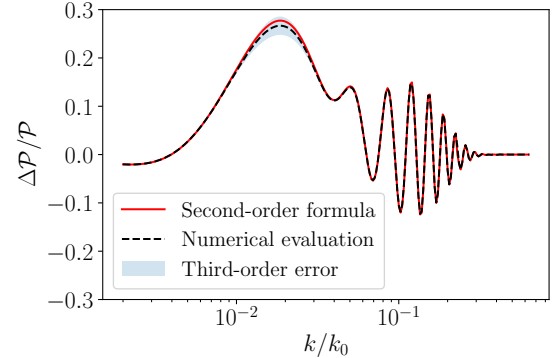

Figure 2: The dotted/dashed lines are the power spectra obtained via the 1st/1st+2nd-order expressions (14) and (18) respectively. The red line is the numerical result.

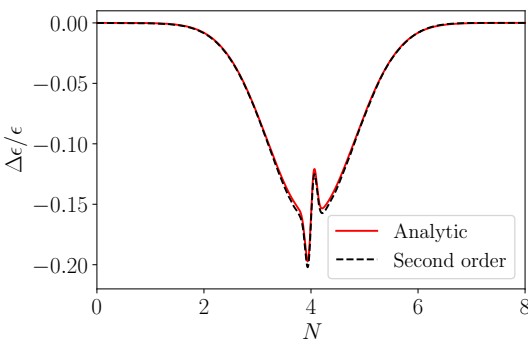

Figure 3: The superposed dashed line is the reconstructed $\Delta\epsilon/\epsilon$ obtained from evaluating (15) using the exact power spectrum in (19).

Figure 4: Numerical evaluation of the power spectrum using the reconstructed $\Delta\epsilon/\epsilon$ shown in Fig. 3, which reproduces the exact power spectrum to within an accuracy of $(\Delta\mathcal{P}_{\mathcal{R}}/\mathcal{P}_{\mathcal{R}})^3$.

(14) is plotted as a dotted (grey) line in Fig. 2, and the 2nd-order correction (18) is plotted as a dashed (black) line. As expected, the analytic expression at 2nd-order everywhere matches the exact power spectrum to within an error of $(\Delta\mathcal{P}_{\mathcal{R}}/\mathcal{P}_{\mathcal{R}})^3$, i.e. to the per cent level for the particular feature model considered here. In Fig. 3, we superpose the result of the reconstructed $\epsilon$ from the exact featureful power spectrum using (19) and (15) with the known analytic one (21). This reconstructed $\epsilon$ is able to reproduce the original fractional change of the power spectrum to within an error of $(\Delta\mathcal{P}_{\mathcal{R}}/\mathcal{P}_{\mathcal{R}})^3$. It is not therefore surprising to see it almost exactly match the original analytic form for $\Delta\epsilon/\epsilon$. When one numerically obtains the power spectrum from the reconstructed $\Delta\epsilon/\epsilon$, we see in Fig. 4 that indeed, it reproduces the original power spectrum to within the appropriate accuracy. Having convinced ourselves that the formalism works as advertised, we now turn to the problem of inverting for the parameters of the EFT of inflation using reconstructed power spectra extracted from CMB data.

## 3 Reconstructing the EFT parameters

Having established the relation between the EFT parameters, $\Delta\epsilon/\epsilon(\tau)$, and $u(\tau) = 1/c_s^2(\tau)-1$, and the induced fractional change in the PPS, $\Delta\mathcal{P}_{\mathcal{R}}/\mathcal{P}_{\mathcal{R}}(k)$, we can now reconstruct the EFT parameter given estimates of the fractional PPS. The starting point will be a previously estimated PPS and its uncertainty, though it is described in Appendix D.2 how the estimation of the PPS can be circumvented entirely. Two possible approaches to the estimation, or reconstruction, of the EFT parameters from the fractional PPS are possible.

One approach is to use the inverse relations mapping $\Delta\mathcal{P}_{\mathcal{R}}/\mathcal{P}_{\mathcal{R}}(k)$ to $X(\tau)$, i.e. (13) and (15) and thereby transform the estimated PPS $\mathcal{P}_{\text{rec}}(k)$ into an estimate of $X(\tau)$. This approach will be adopted here. However, since the estimated PPS, being a reconstruction from noisy data, will be jagged, the estimate of $X(\tau)$ will be so, too.

Another strategy would have been to maximise the likelihood associated with the PPS with respect to $X(\tau)$. The relation between $X(\tau)$ and the PPS it induces is given by the forward relations (12) and (14). The likelihood compares the induced PPS to the PPS estimated from observations, weighting the discrepancy by the PPS covariance matrix. A penalty on the roughness of $X(\tau)$ is then added to the likelihood to select a realistic solution. This is discussed in detail in Appendix D.

The starting point of our work is a previously estimated PPS. It was recovered from the Planck Public Release 2 temperature and polarisation data using Tikhonov regularisation penalising first-order derivatives[9] of the PPS, as explained in detail in [31]. The Planck $TT$, $TE$ and $EE$ likelihood function consists of a pixel-based component for multpoles $\ell \leq 29$ and a Gaussian pseudo-$C_\ell$ component for $30 \leq \ell \leq 2508$. The fractional PPS $\Delta\mathcal{P}_{\mathcal{R}}/\mathcal{P}_{\mathcal{R}}(k)$ was then constructed by subtracting the reconstructed PPS $\mathcal{P}_{\mathcal{R}}(k)$ from the power-law PPS $\mathcal{P}_{\mathcal{R}}^{\text{pow}}(k)$ and dividing by the latter. The PPS and its uncertainty was estimated on a grid of 1900 wave numbers from $k_{\text{min}} = 6 \times 10^{-6}\,\text{Mpc}^{-1}$ to $k_{\text{max}} = 0.75\,\text{Mpc}^{-1}$.

## 4 Reconstructing the inflaton potential

As reviewed in Appendix B, it is possible to reconstruct a potential that would reproduce an arbitrary time varying profile for $\epsilon$ assuming a canonical kinetic term for the inflaton.[10] We caution that this is not the same problem as reconstructing the action for the inflaton background in general, since as discussed in §II, there will be many background models that project onto the same Wilson functions of the EFT of the adiabatic mode and thus many degeneracies exist (cf. [63–66]). Our goal here is to furnish a simple representative from the equivalence class of models that would reproduce any given profile for $\epsilon(\tau)$. From (57), the field profile is

$$\phi(N) = \phi_0 \pm M_{\text{pl}} \int_{N_*}^{N} \mathrm{d}N' \sqrt{2\epsilon(N')}, \tag{23}$$

where the choice $\pm$ corresponds to whether we want the inflaton (and the potential it descends in) to move towards increasing or decreasing values of $\phi$. The potential can correspondingly

---

[9]More precisely, the penalty is proportional to $\int_0^\infty \mathrm{d}\log k \, (\mathrm{d}\log\mathcal{P}_{\mathcal{R}}/\mathrm{d}\log k - (n_s - 1))^2$ where departures from a power-law $\propto k^{n_s-1}$ with spectral index $n_s$ are penalised.

[10]It is also possible to reproduce this procedure given an *a priori* fixed non-canonical form of the kinetic term. This is one of the many model degeneracies inherent in our procedure. However, since goal of the present exercise is merely to write down a simple representative model, assuming a canonical form is sufficient for our purposes.

be reconstructed through (58):

$$V(N) = V(N_*)\exp\left[-\frac{1}{3}\int_{N_*}^{N} dN'\left(\frac{d\epsilon}{dN'} + 6\epsilon\right)\right].$$ (24)

Inverting for $\phi$ as a function of $N$ and substituting into the potential above results in $V(\phi)$.

Before turning our attention towards explicit reconstructions from CMB data, we make a quick detour to discuss how one could obtain any given reconstructed PPS with a variation in the speed of sound. We note that one could just have straightforwardly inserted the expression (19) into (13) to find the reconstructed $c_s^2$ as a function of time, however it turns out that when one does so for both $\Lambda$CDM and EdS around an attractor for which $c_s = 1$, one necessarily requires transient phases of $c_s > 1$. One can evade this by requiring that the attractor be such that it has some constant $c_0 < 1$ (cf. [61]), in which case the relevant inversion formula is given by:

$$\frac{1}{c_s^2} - \frac{1}{c_0^2} = \frac{1}{\pi}\int_{-\infty}^{0}\frac{dk}{k}\frac{\Delta_1\mathcal{P}_\mathcal{R}}{\mathcal{P}_\mathcal{R}}(k)\sin(-2kc_0\tau).$$ (25)

It should not come as a surprise that there are many ways to obtain the same PPS from different choices for the functional parameters of the EFT of inflation, and the above is a manifestation of this degeneracy (see also [67–70] for a discussion of dualities between different backgrounds that produce the same PPS). An analysis of whether CMB data shows evidence for variations in the sound speed have been done within a 1st-order formalism [71,72], and our formalism to invert for $c_s$ readily applies to this case as well. However, as discussed in §II, reductions in $c_s$ are sourced by operators that are at least two degrees higher in derivatives than those that source changes in $\epsilon$, and so if our goal is to look for the simplest representative background models that can reproduce any given reconstructed features, it is reasonable to restrict to features induced by variations in $\epsilon$.

## 5 Results for $\Lambda$CDM

The PPS estimated from Planck Release 2 data *assuming* a $\Lambda$CDM model consistent with the best-fit Planck Release 2 parameters is shown in Fig. 5 including estimated Bayesian and frequentist uncertainties and a fiducial power-law PPS with spectral index $n_s = 0.968$. There are few indications of departures from a power-law PPS when the best-fit $\Lambda$CDM cosmological model is assumed. The most notable deviation is near $k \sim 2 \times 10^{-3}\,\mathrm{Mpc}^{-1}$ which receives dominant contributions from multipoles $\ell \sim 28$.

The reconstruction of $\epsilon(\tau)$ shown in Fig. 6 derived from this PPS is normalised such that the pivot scale $k_* = 2\times10^{-3}\,\mathrm{Mpc}^{-1}$ exits the horizon at $N = 0$ e-folds. An attractor background slow-roll parameter $\epsilon = 10^{-4}$ was assumed.

The reconstruction displays a prominent peak around $N \sim 3.5$ e-folds due to the $\ell \sim 28$ feature. The $1\sigma$ confidence interval on the reconstruction is given by the square root of the diagonal elements of its associated covariance matrix, obtained as described in Appendix A.

On the plot two error bands are shown, one confidence interval derived considering only the diagonal elements of the frequentist covariance matrix which describes the error in the reconstructed PPS, and the other considering the full matrix. These bands only indicate the trend in the error band as a complete analysis would require evaluating the full likelihood. In the diagonal approximation the statistical significance of a feature may appear to be high, but including the full covariance matrix increases the uncertainty in the reconstruction and lowers the significance. This is essentially because of cosmic variance on large scales which propagates to intermediate scales due to correlations between nearby wave numbers. Moreover the EFT

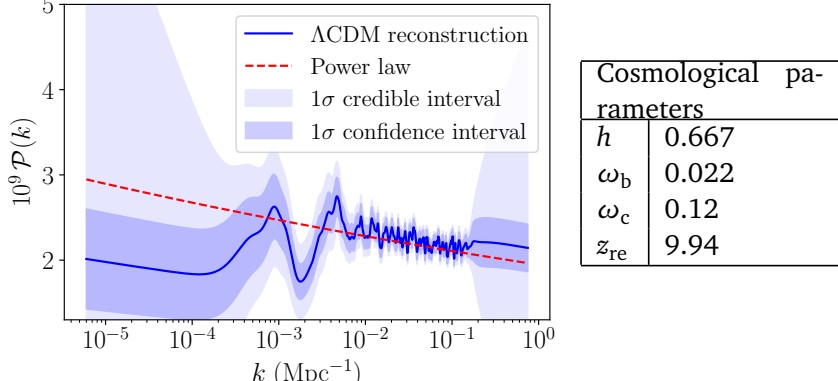

Figure 5: Reconstruction (blue line) of the PPS from Planck Public Release 2 $TT$, $TE$ and $EE$ data assuming a $\Lambda$CDM cosmological model with cosmological parameters listed in the table (right). The purple band indicates the $1\sigma$ confidence interval and the light blue band indicates the $1\sigma$ credible interval. A power-law PPS (red dashed line) with $n_s = 0.968$ is superimposed.

parameters are non-local functions of the PPS, so they receive contributions from a range of wave numbers with finite support. However, it is beyond the scope of this work to present a full statistical analysis, our aim here being to demonstrate accurate EFT parameter reconstruction from a cosmological data set.

Using (55) and (24) we obtain the potential $V(\phi)$ corresponding to the reconstructed $\epsilon$ for $\Lambda$CDM, which is shown in Fig. 7. The first thing to note is that the potential itself appears not dissimilar to that produced by a smooth polynomial. However the derivatives of the potential exhibit fine scale features, whose purpose is to knock the inflaton off the attractor solution as it evolves (right panel, Fig. 7). As expected, the derivatives of the potential closely track the reconstructed $\epsilon$ since the potential definition of the slow roll parameter $\epsilon_V \equiv M_{\rm pl}^2 (\partial_\phi V/V)^2$ tends to the Hubble hierarchy definition $\epsilon = -\dot{H}/H^2$ when $\epsilon \ll 1$. One might reasonably ask how such effective potentials could be produced from an underlying parent theory. We shall detail various possibilities in our concluding discussion.

## 6 Results for Einstein-de Sitter

The same procedure, reconstructing the PPS from the Planck Public Release 2 data, was repeated for a cosmology *without* dark energy, the flat EdS cold+hot dark matter (CHDM) model. As shown earlier [36,37] it requires a Hubble constant of $h \simeq 0.44$ and a 12% hot dark matter component of neutrinos with $\sum m_\nu = 2.2$ eV. As seen in Fig. 8, large features in the reconstructed PPS are necessary for the EdS cosmology to match the data. These consist of a bump around $k \sim 2 \times 10^{-2}$ Mpc$^{-1}$ followed by oscillations that continue until $k \sim 2 \times 10^{-1}$ Mpc$^{-1}$. These oscillations ensure that the model fits the small scale CMB acoustic peaks. A model involving two successive phase transitions during multiple inflation which reproduces the general shape of the reconstructed PPS had been proposed in [36,37], however it admittedly does not yield the oscillatory small-scale fine structure.

The EdS $\epsilon(\tau)$ estimate of Fig. 10 again exhibits a large peak at $N \sim 3.5$ but now also features seemingly sharp oscillations at $N \sim 5$ corresponding to the small scale oscillations in the PPS. Repeating the same error analysis as was done for the $\Lambda$CDM case, the error in the reconstructed EFT parameter due to the uncertainty in the estimated PPS was obtained. Both

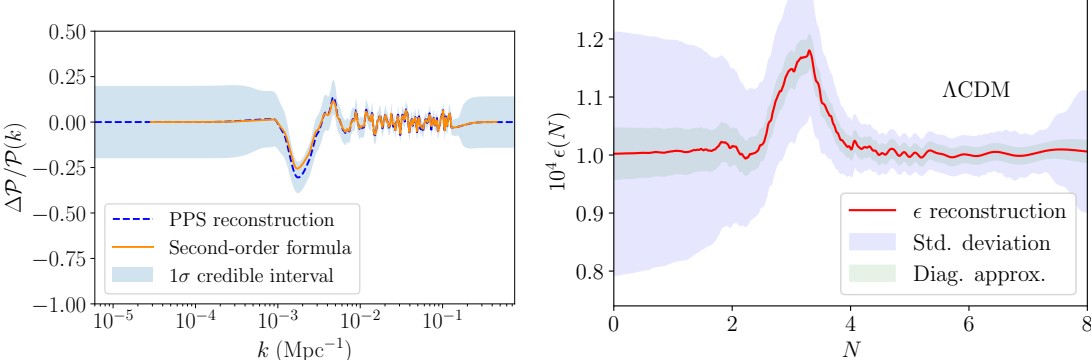

Figure 6: The right panel shows the 2nd-order reconstructed $\epsilon$ for the $\Delta\mathcal{P}_{\mathcal{R}}/\mathcal{P}_{\mathcal{R}}$ estimated from Planck data assuming $\Lambda$CDM (left panel, dashed blue line). The blue (full covariance matrix) and green (its diagonal approximation) shaded bands indicate the $1\sigma$ uncertainties in $\epsilon$ due to errors in the estimated PPS. The orange line in the left panel is the PPS obtained by numerical integration of the reconstructed $\epsilon$.

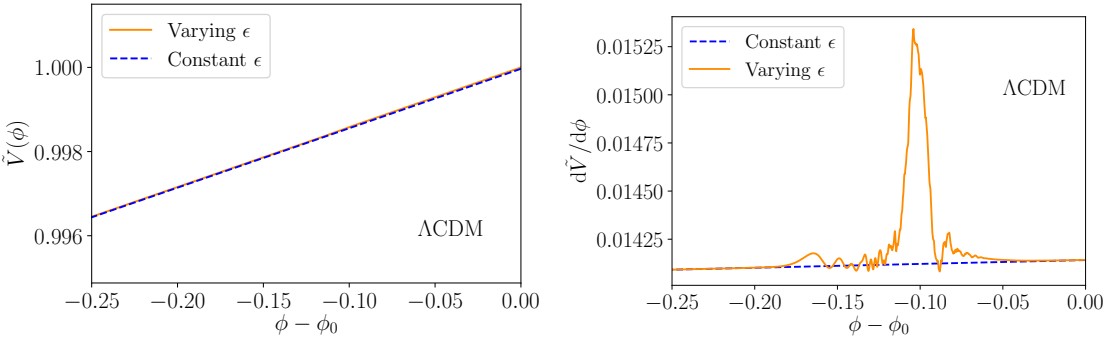

Figure 7: The left panel shows the potential $\tilde{V} = V(\phi)/V(\phi_0)$ corresponding to the reconstructed $\Delta\epsilon/\epsilon$ superposed on the attractor potential (dashed blue line) – the right panel is its derivative.

the full and diagonal contributions of the PPS covariance matrix to the standard deviation of $\epsilon$ were again considered. It is seen that the off-diagonal elements make a large contribution to the uncertainty in $\epsilon$ and lower the statistical significance of the features. However the sharp feature at $N \sim 5$ is still required when an EdS cosmology is assumed.

Although this may seem like a sudden change in an EFT parameter over $< 1$ e-fold, the degree of suddenness is quantified by the second term in the Hubble hierarchy $\eta \equiv \dot{\epsilon}/\epsilon H$, which is bounded throughout by $|\eta| \lesssim 1.5$, leaving us safely within the single clock regime [73] (also true for the $\Lambda$CDM case (Fig. 7)). As in the previous section, one can reconstruct the potential that could have given rise to the reconstructed feature that best fits an underlying EdS cosmology (cf. Fig. 9). We see again that the potential itself looks similar to a smooth polynomial over the field excursion needed to produce the observed modes. However, its derivatives vary along the trajectory tracking $\epsilon$ closely in just such a manner as to knock the background off the attractor, producing the required features. This occurs in a manner that produces a fit to the reconstructed PPS accurate to the percent level *without* needing to invoke any phase transitions (as in [36,37]). It remains for us to elaborate on the nature of the parent theory that could have produced such an effective potential.

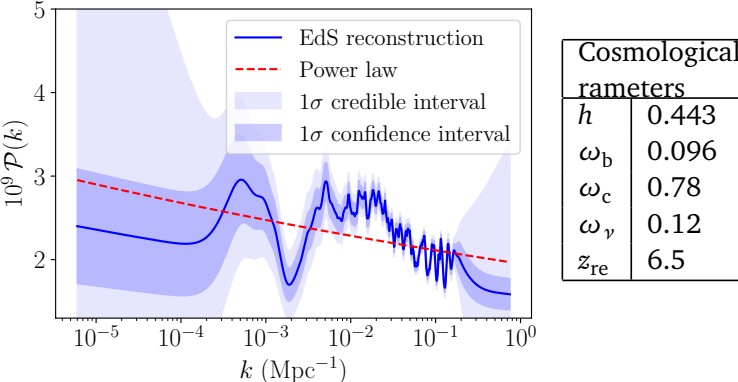

Figure 8: The estimated PPS for the EdS cosmological model with neutrino dark matter from Planck Release 2 $TT$, $TE$ and $EE$ data. The left panel shows the reconstructed PPS (blue line) with credible (purple band) and confidence intervals (light blue band) with $n_s = 0.968$ power-law PPS (red dashed line) superimposed. The right panel shows the cosmological parameters.

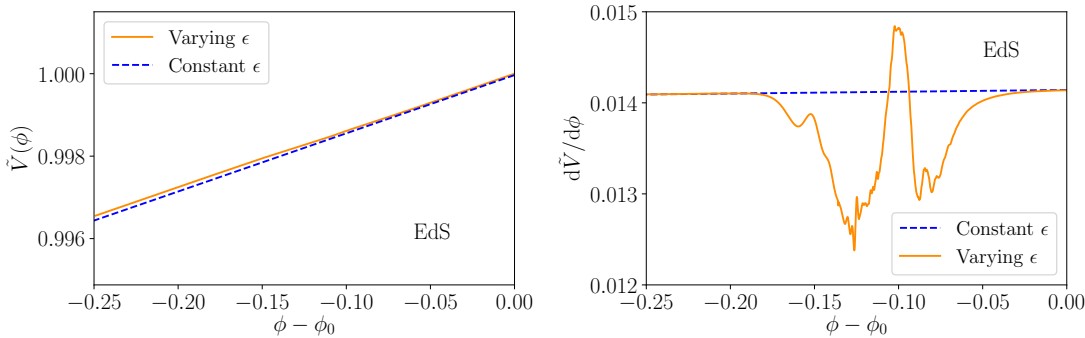

Figure 9: The left panel shows the potential $\tilde{V} = V(\phi)/V(\phi_0)$ corresponding to $\Delta\epsilon/\epsilon$ superposed on the attractor potential (dashed blue line), and its derivative (right panel).

## 7 Discussion

Having seen how to reconstruct potentials that can produce any given power spectrum, one might wonder how such effective potentials might arise in realistic settings. Viewing the effective action for the inflaton background as having been obtained by integrating out all heavy degrees of freedom in the parent theory, one can for example obtain leading order (adiabatic) corrections to the inflaton potential of the form (cf. (63) and (71))

$$\partial_\phi V_{\text{CW}}(\phi) = \frac{\partial_\phi M^2(\phi)}{32\pi^2} M^2(\phi) \ln\left[ M^2(\phi)/\mu^2 \right], \tag{26}$$

where the above was obtained by integrating out a heavy field with an effective mass given by $M^2(\phi)$ that is taken to vary weakly enough with respect to $\phi$ i.e.

$$\dot{\phi}_0 \partial_\phi M/M^2 \ll 1, \tag{27}$$

where $\phi_0$ denotes the background trajectory, and where the inflaton effective potential is given by the sum of the above correction plus the background field contribution $V_{\text{eff}} = V_{\text{inf}} + V_{\text{CW}}$

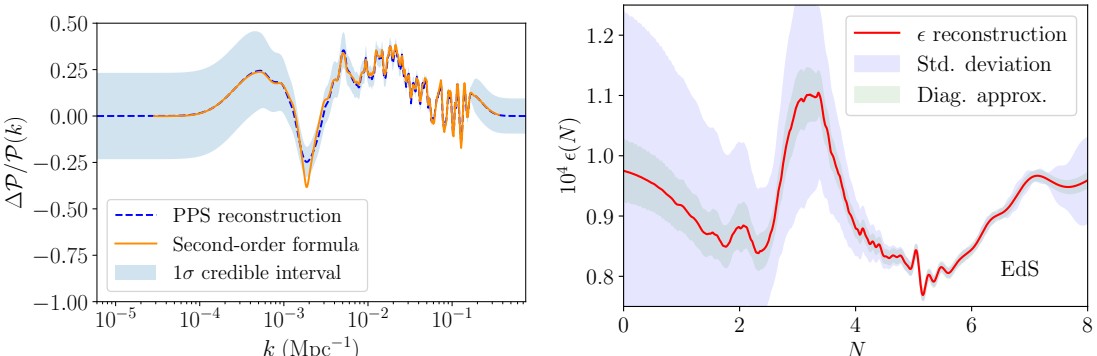

Figure 10: The right panel shows the 2nd-order reconstructed $\epsilon$ for the $\Delta\mathcal{P}_\mathcal{R}/\mathcal{P}_\mathcal{R}$ (left panel, dashed blue line) estimated from Planck data assuming the EdS cosmological model. The green (diagonal approximation) and blue (full matrix) bands indicate the $1\sigma$ uncertainties in $\epsilon$ due to errors in the estimated PPS. The orange line is the result of numerical integration of the power spectrum given the reconstructed $\epsilon$.

(cf. eq. 63). Violating (27) necessarily implies particle production resulting from higher orders in the adiabatic expansion that one can calculate (reviewed in Appendix C). Indeed, the possibility of localised particle production events along the inflaton trajectory was considered in [74–78] and can generate additional features in the effective potential. However one has to study these cases more carefully given the possibility of production and subsequent decay of isocurvature modes – removing us from the single clock context upon which this study relies. One can nevertheless quantify the requirement of staying within the adiabatic approximation in generating the features in the effective potential required to produce the finer features, such as in the EdS case (cf. Fig. 10). Re-expressing (27) as

$$\sqrt{2\epsilon}\frac{H}{M}\frac{\partial}{\partial\tilde{\phi}}\log M = \sqrt{\frac{\epsilon}{8}}\frac{H}{M}\frac{\partial}{\partial\tilde{\phi}}\log V_{\rm CW} \ll 1 \,, \tag{28}$$

where the partial derivative is with respect to $\tilde{\phi} = \phi/M_{\rm pl}$, we find that the logarithmic derivative of the potential around the finer feature in Fig. 9 to be order unity. Given the value of $\epsilon$ around the fiducial attractor presented in the plot is $\epsilon_0 = 10^{-4}$, and given the assumption that the mass of the heavy field is much greater than Hubble, the condition (28) is readily satisfied.

We stress that the formalism developed here allows one to obtain the parameters of the EFT of inflation given any particular set of assumptions for the reconstruction. The examples presented here were of reconstructions presuming a background $\Lambda$CDM or Einstein-de Sitter cosmology with a fixed set of parameters, but are equally applicable to other examples. [11] One can thus 'invert' for background models that could reproduce any given reconstructed primordial power spectrum provided $(\Delta\mathcal{P}_\mathcal{R}/\mathcal{P}_\mathcal{R})^3$ is less than the $1\sigma$ confidence interval of the reconstruction $\Sigma(k)$.

---

[11]For instance, one can consider the possibility that discrepancy between low redshift measurements of $H_0$ and those obtained from CMB observations can be projected onto a primordial power spectrum with specific features [79].

## Acknowledgements

We would like to thank Ana Achucarro and Gonzalo Palma for helpful discussions and the Referee for a helpful query which led us to critically reexamine the error bands on our reconstructed parameters. The authors were supported by a Niels Bohr Professorship award to SS by the Danmarks Grundforskningsfond.

## A  Feature inversion

We recall the leading order action (8):

$$S_2 = M_{\mathrm{pl}}^2 \int d^4x \, a^3 \epsilon \left( \frac{\dot{\mathcal{R}}^2}{c_{\mathrm{s}}^2} - \frac{(\partial \mathcal{R})^2}{a^2} \right). \tag{29}$$

We imagine the background of interest (characterised by $c_{\mathrm{s}}(\tau)$ and $\epsilon(\tau)$) is a small perturbation about a fiducial attractor solution with constant $\epsilon$ and $c_{\mathrm{s}} = 1$, to which it tends at early and late times. The small quantities $\Delta\epsilon/\epsilon(\tau)$ and $u(\tau) = 1/c_{\mathrm{s}}(\tau) - 1$ then define a perturbative expansion. We use the in-in formalism to calculate the fractional change in the power spectrum at 1st- and 2nd-order in $\Delta\epsilon/\epsilon(\tau)$ and $u(\tau)$. Some of the details not elaborated upon in this appendix can be found in [80].

The interaction Hamiltonian for a feature induced by a change in $c_{\mathrm{s}}$ (with $\epsilon$ held fixed) can be read off from (29) as

$$H_{\mathrm{int}} = \epsilon M_{\mathrm{Pl}}^2 \int d^3x \, a(\tau)^2 u(\tau) (\mathcal{R}'(\tau))^2, \tag{30}$$

and the corresponding interaction Hamiltonian for a feature induced by a change in $\epsilon$ (keeping $c_{\mathrm{s}}$ fixed) as

$$H_{\mathrm{int}} = \epsilon M_{\mathrm{Pl}}^2 \int d^3x \, a^2(\tau) \frac{\Delta\epsilon}{\epsilon}(\tau) \left( -(\mathcal{R}')^2 + (\partial_i \mathcal{R})^2 \right). \tag{31}$$

The curvature perturbation is expanded in Fourier modes as

$$\mathcal{R}(\tau) = \int \frac{d^3k}{(2\pi)^3} \left( \hat{a}_{\mathbf{k}} \mathcal{R}_{\mathbf{k}}(\tau) e^{i\mathbf{k}\cdot\mathbf{x}} + \hat{a}_{\mathbf{k}}^\dagger \mathcal{R}_{\mathbf{k}}^*(\tau) e^{-i\mathbf{k}\cdot\mathbf{x}} \right), \tag{32}$$

and quantised with the commutation relation $[\hat{a}_{\mathbf{k}}, \hat{a}_{\mathbf{k}'}^\dagger] = (2\pi)^3 \delta^{(3)}(\mathbf{k} + \mathbf{k}')$. The fiducial attractor defines the 'free field' mode functions

$$\mathcal{R}_k(\tau) = \frac{iH}{2M_{\mathrm{Pl}}\sqrt{\epsilon k^3}} (1 + ik\tau) e^{-ik\tau}. \tag{33}$$

The perturbation of order $n$ to the two-point correlation function of curvature fluctuations at time $\tau$ induced by a feature is given by

$$\Delta_n \langle \mathcal{R}_{\mathbf{k}}(\tau) \mathcal{R}_{\mathbf{k}'}(\tau) \rangle = i^n \int_{-\infty}^{\tau} d\tau_n \int_{-\infty}^{\tau_n} d\tau_{n-1} \cdots$$
$$\int_{-\infty}^{\tau_2} d\tau_1 \left\langle [H_{\mathrm{int}}(\tau_1), \cdots [H_{\mathrm{int}}(\tau_2), [H_{\mathrm{int}}(\tau_n), \mathcal{R}_{\mathbf{k}}(\tau) \mathcal{R}_{\mathbf{k}'}(\tau)]]] \right\rangle_0 . \tag{34}$$

Here the subscript on the expectation value denotes that the quantities appearing in the expectation value are free field interaction picture operators evaluated in the adiabatic (Bunch-Davies) vacuum[12]. The dimensionless power spectrum is related to the two-point correlation function by

$$(2\pi)^3 \delta^{(3)}(\mathbf{k}+\mathbf{k}')\mathcal{P}_{\mathcal{R}}(k) = \frac{k^3}{2\pi^2}\langle\mathcal{R}_{\mathbf{k}}(0)\mathcal{R}_{\mathbf{k}'}(0)\rangle\,, \tag{35}$$

which is evaluated at the end of inflation when $\tau = 0$. Substituting the Hamiltonian (30) in (34) and using (33) gives

$$\frac{\Delta\mathcal{P}_{\mathcal{R}}}{\mathcal{P}_{\mathcal{R}}}(k) = -k\int_{-\infty}^{0} d\tau\, u(\tau)\sin(2k\tau) \tag{36}$$

for the fractional change in the power spectrum due to a variation in the speed of sound with $\epsilon$ held constant.

From the above, it should be clear that one can simply apply an inverse integral transform to obtain the functions $u(\tau)$ in terms of $\Delta\mathcal{P}_{\mathcal{R}}/\mathcal{P}_{\mathcal{R}}$. In order to implement the inversion, $u(\tau)$ is extended over the entire real line as an odd function of $\tau$ (implying $\Delta\mathcal{P}_{\mathcal{R}}/\mathcal{P}_{\mathcal{R}}$ is an even function of $k$, similarly extending $k$ over the entire real line) and the sin function is written as a sum of exponentials. Then an inverse Fourier transform yields

$$u(\tau) = -\frac{4}{\pi}\int_0^{\infty}\frac{dk}{k}\frac{\Delta\mathcal{P}_{\mathcal{R}}}{\mathcal{P}_{\mathcal{R}}}(k)\sin(2k\tau)\,. \tag{37}$$

With slightly more work, one can show that a time variation in $\epsilon$ results in a feature of the form

$$\frac{\Delta\mathcal{P}_{\mathcal{R}}}{\mathcal{P}_{\mathcal{R}}}(k) = \frac{1}{k}\int_{-\infty}^{0}\frac{d\tau}{\tau^2}\frac{\Delta\epsilon}{\epsilon}(\tau)((1-2k^2\tau^2)\sin(2k\tau)-2k\tau\cos(2k\tau))\,. \tag{38}$$

By again extending the function $\Delta\epsilon/\epsilon$ over the entire real line as an odd function this can be rewritten as

$$\begin{aligned}\frac{\Delta\mathcal{P}_{\mathcal{R}}}{\mathcal{P}_{\mathcal{R}}}(k) &= \frac{1}{2k}\int_{-\infty}^{\infty}\frac{d\tau}{\tau^2}\frac{\Delta\epsilon}{\epsilon}(\tau)\left(-i+i2k^2\tau^2-2k\tau\right)e^{2ik\tau}\\ &= \frac{1}{2k}\int_{-\infty}^{\infty}d\tau\frac{\Delta\epsilon}{\epsilon}(\tau)i\left(-\frac{1}{\tau^2}-\frac{1}{2}\frac{\partial^2}{\partial\tau^2}+\frac{1}{\tau}\frac{\partial}{\partial\tau}\right)e^{2ik\tau}\,.\end{aligned} \tag{39}$$

Since by assumption that $\Delta\epsilon/\epsilon$ vanishes asymptotically an integration by parts results in

$$\frac{\Delta\mathcal{P}_{\mathcal{R}}}{\mathcal{P}_{\mathcal{R}}}(k) = \frac{1}{2k}\int_{-\infty}^{\infty}d\tau\, e^{2ik\tau}i\left\{\left(-\frac{1}{2}\frac{d^2}{d\tau^2}-\frac{1}{\tau}\frac{d}{d\tau}\right)\frac{\Delta\epsilon}{\epsilon}(\tau)\right\}\,. \tag{40}$$

Performing the inverse Fourier transform gives the following inhomogeneous differential equation for $\Delta\epsilon/\epsilon$

$$\left(\frac{d^2}{d\tau^2}+\frac{2}{\tau}\frac{d}{d\tau}\right)\frac{\Delta\epsilon}{\epsilon}(\tau) = \frac{4i}{\pi}\int_{-\infty}^{\infty}k\,dk\, e^{-2ik\tau}\frac{\Delta\mathcal{P}_{\mathcal{R}}}{\mathcal{P}_{\mathcal{R}}}(k)\,. \tag{41}$$

---

[12]When working to 2nd-order, one has to be mindful of an important subtlety as to how one selects the correct vacuum, with the formal equivalence between the in-in correlation function and the expression (34) no longer valid when one deforms the contour of integration to pick up a small imaginary part in the infinite past [81]. However this difference manifests only when calculating loops

This can be solved by factoring the differential operator on the right hand side as

$$\frac{1}{\tau^2}\frac{\mathrm{d}}{\mathrm{d}\tau}\left(\tau^2\frac{\mathrm{d}}{\mathrm{d}\tau}\right)\frac{\Delta\epsilon}{\epsilon}(\tau) = \frac{4i}{\pi}\int_{-\infty}^{\infty}k\,\mathrm{d}k\,e^{-2ik\tau}\frac{\Delta\mathcal{P}_\mathcal{R}}{\mathcal{P}_\mathcal{R}}(k). \tag{42}$$

Evidently, any kernel satisfying the inhomogeneous equation

$$\frac{1}{\tau^2}\frac{\mathrm{d}}{\mathrm{d}\tau}\left(\tau^2\frac{\mathrm{d}}{\mathrm{d}\tau}\right)g(\tau,k) = e^{-2ik\tau}, \tag{43}$$

whose solution consistent with the boundary conditions imposed on $\Delta\epsilon/\epsilon$ is

$$g(\tau,k) = \frac{e^{-2ik\tau}}{4k^2}\left(-1+\frac{i}{k\tau}\right) - 1 - \frac{i}{k\tau}, \tag{44}$$

and can be used to obtain $\Delta\epsilon/\epsilon$ as a functional of $\Delta\mathcal{P}_\mathcal{R}/\mathcal{P}_\mathcal{R}$. Convolving the above with the integrand of (42) finally gives

$$\begin{aligned}
\frac{\Delta\epsilon}{\epsilon}(\tau) &= \frac{2}{\pi}\int_0^\infty\frac{\mathrm{d}k}{k}\frac{\Delta\mathcal{P}_\mathcal{R}}{\mathcal{P}_\mathcal{R}}(k)\left(\frac{1-\cos(2k\tau)}{k\tau}-\sin(2k\tau)\right) \\
&= \frac{2}{\pi}\int_0^\infty\frac{\mathrm{d}k}{k}\frac{\Delta\mathcal{P}_\mathcal{R}}{\mathcal{P}_\mathcal{R}}(k)\left(\frac{2\sin^2(k\tau)}{k\tau}-\sin(2k\tau)\right).
\end{aligned} \tag{45}$$

An expression for the 1st-order fractional change in the power spectrum when $c_s$ and $\epsilon$ vary *simultaneously* can be found in [46].

It is a straightforward if slightly tedious exercise to calculate the feature induced by a change in the speed of sound to 2nd-order in perturbation theory. After some manipulation it can be expressed in the remarkably simple form (see [80] for details)

$$\frac{\Delta_2\mathcal{P}_\mathcal{R}}{\mathcal{P}_\mathcal{R}}(k) = k^2\left(\int_{-\infty}^0\mathrm{d}\tau_1 u(\tau_1)\sin(2k\tau_1)\right)^2. \tag{46}$$

Comparing this to the 1st-order result (36), we thus see that

$$\left.\frac{\Delta_2\mathcal{P}_\mathcal{R}}{\mathcal{P}_\mathcal{R}}(k)\right|_{c_s} = \left(\frac{\Delta_1\mathcal{P}_\mathcal{R}}{\mathcal{P}_\mathcal{R}}(k)\right)^2_{c_s}. \tag{47}$$

Proceeding similarly, one can show for a non-zero $\Delta\epsilon/\epsilon$ that the 2nd-order feature that results is given by

$$\begin{aligned}
\frac{\Delta_2\mathcal{P}_\mathcal{R}}{\mathcal{P}_\mathcal{R}}(k) = -2\int_{-\infty}^0\frac{\mathrm{d}\tau_2}{\tau_2^2}\frac{\Delta\epsilon}{\epsilon}(\tau_2)\int_{-\infty}^{\tau_2}\frac{\mathrm{d}\tau_1}{\tau_1^2}\frac{\Delta\epsilon}{\epsilon}(\tau_1)&\left\{k^2\tau_1^2\tau_2^2\,\mathrm{Im}\left[e^{ik(\tau_2-2\tau_1)}\right]\mathrm{Im}\left[e^{-ik\tau_2}\right]\right. \\
&+ k^{-2}\,\mathrm{Im}\left[e^{ik(\tau_2-2\tau_1)}(1-ik\tau_1)(1+ik\tau_2)^2\right]\mathrm{Im}\left[e^{-ik\tau_2}(1+ik\tau_2)\right] \\
&- \tau_1^2\,\mathrm{Im}\left[e^{ik(\tau_2-2\tau_1)}(1-ik\tau_2)\right]\mathrm{Im}\left[e^{-ik\tau_2}(1+ik\tau_2)\right] \\
&\left.- \tau_2^2\,\mathrm{Im}\left[e^{ik(\tau_2-2\tau_1)}(1+ik\tau_1)^2\right]\mathrm{Im}\left[e^{-ik\tau_2}\right]\right\}.
\end{aligned}$$

As done for $u(\tau)$, by extending $\Delta\epsilon/\epsilon$ as an odd function over the entire real line the factor in the inner most integrand can be expressed as

$$\frac{\Delta\epsilon}{\epsilon}(\tau_1) = \int_{-\infty}^{\infty}\frac{d\omega}{2\pi}h(\omega)e^{i\omega\tau_1}, \tag{48}$$

where $h(\omega)$ is also an odd function. Interchanging the $\tau_1$ and $\omega$ integrals, performing the $\tau_1$ integration explicitly and then performing a contour integral for $\omega$ results in the intermediate expression

$$\frac{\Delta_2 \mathcal{P}_\mathcal{R}}{\mathcal{P}_\mathcal{R}}(k) = \int_{-\infty}^{\infty} \frac{d\tau_2}{2\tau_2^2} \frac{\Delta\epsilon}{\epsilon}(\tau_2) h(2k) e^{-2ik\tau} \left(1 + 2ik\tau_2 - 2k^2\tau_2^2\right), \tag{49}$$

where the simplification is due to the fact that various terms do not contribute residues, and that the only poles of the integrand are at $\omega = \pm 2k$. Recognising the combination in the parentheses from (39), and evaluating $h(2k)$ as the Fourier transform of (45) finally results in

$$\left.\frac{\Delta_2 \mathcal{P}_\mathcal{R}}{\mathcal{P}_\mathcal{R}}(k)\right|_\epsilon = \left(\frac{\Delta_1 \mathcal{P}_\mathcal{R}}{\mathcal{P}_\mathcal{R}}(k)\right)_\epsilon^2. \tag{50}$$

Therefore for features induced by variations in either $c_s$ or $\epsilon$, we see from summing the 1st- and 2nd-order corrections that the induced fractional change in the power spectrum is given by the simple expression

$$\frac{\Delta \mathcal{P}_\mathcal{R}}{\mathcal{P}_\mathcal{R}}(k) = \frac{\Delta_1 \mathcal{P}_\mathcal{R}}{\mathcal{P}_\mathcal{R}}(k) + \left(\frac{\Delta_1 \mathcal{P}_\mathcal{R}}{\mathcal{P}_\mathcal{R}}(k)\right)^2 + \ldots, \tag{51}$$

where the ellipses denote terms from third order in perturbation theory that could possibly include logarithmic derivatives of the 1st-order fractional change e.g. $(\frac{\Delta_1 \mathcal{P}_\mathcal{R}}{\mathcal{P}_\mathcal{R}})^3 + (\frac{\Delta_1 \mathcal{P}_\mathcal{R}}{\mathcal{P}_\mathcal{R}})^2 \frac{d}{d\log k} \frac{\Delta_1 \mathcal{P}_\mathcal{R}}{\mathcal{P}_\mathcal{R}}$ etc. Neglecting these for now, we can invert any given reconstruction for the functional parameters in the EFT of the adiabatic mode up to 2nd-order by inserting the reconstructed power spectrum into the left hand side of the above and solving for $\Delta_1 \mathcal{P}_\mathcal{R}/\mathcal{P}_\mathcal{R}$. The result is

$$\frac{\Delta_1 \mathcal{P}_\mathcal{R}}{\mathcal{P}_\mathcal{R}}(k) = \frac{1}{2}\left(-1 + \sqrt{1 + 4\frac{\Delta\mathcal{P}_{\mathrm{rec}}}{\mathcal{P}_\mathcal{R}}(k)}\right), \tag{52}$$

where the input reconstructed power spectrum is denoted $\Delta\mathcal{P}_{\mathrm{rec}}/\mathcal{P}_\mathcal{R}$. Employing (52) as the input fractional change in the power spectrum in either (37) or (45) implies that the profiles thus obtained for $c_s$ or $\epsilon$ will reproduce the reconstructed power spectrum up to the accuracy of the terms neglected in (51), that is, to order $(\Delta_1 \mathcal{P}_\mathcal{R}/\mathcal{P}_\mathcal{R})^3$. Thus within our 2nd-order formalism, one can invert for parameters of the EFT of inflation *that can reproduce reconstructed features as large as 25% with roughly two percent precision*, as demonstrated in Fig. 2.

We return now to the limits of validity of the expression (52), which presumed the negligibility of the third order corrections. We first observe that the 2nd-order inversion cannot hold for features that dip below $\Delta_1 \mathcal{P}_\mathcal{R}/\mathcal{P}_\mathcal{R} = -1/4$ since the argument of the square root becomes imaginary below this, meaning that the higher order terms in the expansion are needed in order to extract a real root. Furthermore, we note that the precision with which we are obliged to calculate is such that any inferred EFT parameters must reproduce the reconstructed feature to within the $1\sigma$ confidence interval $\Sigma(k)$ denoted by light blue bands in Figs. 6 and 10. Hence, one can justify neglecting higher order corrections whenever it is comparable to the $1\sigma$ confidence interval of the reconstructed power spectrum $\Sigma(k)$ since that sets the threshold for the required accuracy of our model inversion. Therefore, provided that the reconstructed feature is such that

$$\left(\frac{\Delta\mathcal{P}_\mathcal{R}}{\mathcal{P}_\mathcal{R}}\right)^3 \lesssim \Sigma(k), \tag{53}$$

then the 2nd-order formalism detailed above suffices, where the left hand side of the above is understood to be a series of terms possibly including terms containing logarithmic derivatives of the 1st-order feature, as discussed below (51).

## A.1 Error analysis

We obtain covariance matrices $\Sigma_\mathbf{u}$ and $\Sigma_\epsilon$ for $u(\tau)$ and $\Delta\epsilon/\epsilon(\tau)$ respectively from $\Sigma_\mathbf{p}$, the covariance matrix for $\Delta\mathcal{P}_{\mathrm{rec}}/\mathcal{P}_\mathcal{R}(k)$. When displayed in plots, the confidence intervals of the $u(\tau)$ and $\Delta\epsilon/\epsilon(\tau)$ reconstructions are given by the square root of the diagonal elements of $\Sigma_\mathbf{u}$ and $\Sigma_\epsilon$. We calculate the confidence intervals of the EFT parameters both neglecting and including the off-diagonal elements of $\Sigma_\mathbf{p}$, displaying both error bands. It should be kept in mind that these confidence intervals only partially account for the uncertainties in the EFT parameters as the parameter values at different e-folds $N$ and $N'$ are correlated.

## B Potential reconstruction from $\epsilon$

We review in this Appendix how to reconstruct a potential given a specified history of $\epsilon$ assuming that the inflaton – denoted $\phi$ – is a minimally coupled, canonical normalised scalar field (see [73] for a similar reconstruction applied to large features of the sort that can generate primordial black holes). We begin with the equation of motion expressed in terms of e-folds $N$ as the time variable

$$H^2 \frac{d^2\phi}{dN^2} + \left(3H^2 + H\frac{dH}{dN}\right)\frac{d\phi}{dN} + \frac{\partial V}{\partial\phi} = 0 \,, \tag{54}$$

equivalent to

$$\frac{d\epsilon}{dN} = -(3-\epsilon)\left[2\epsilon + \frac{d\phi}{dN}\frac{\partial_\phi V}{V}\right] \,, \tag{55}$$

which follows from the definition $2M_{\mathrm{pl}}^2\epsilon = (d\phi/dN)^2$ and the Friedmann equations. Presuming now that $\epsilon \ll 3$, one can approximate the above as

$$\frac{d\epsilon}{dN} = -6\epsilon + \frac{d\log V^{-3}}{dN} \,. \tag{56}$$

The defining equation for $\epsilon$ means that we can straightforwardly reconstruct the field profile given $\epsilon$ as a function of $N$

$$\phi(N) = \phi_* \pm M_{\mathrm{pl}}\int_{N_*}^{N} dN' \sqrt{2\epsilon(N')} \,. \tag{57}$$

Similarly, we can also straightforwardly integrate (56) to obtain

$$V(N_*)\exp\left[-\frac{1}{3}\int_{N_*}^{N} dN'\left(\frac{d\epsilon}{dN'} + 6\epsilon\right)\right] = V(N) \,, \tag{58}$$

yielding $\phi$ and $V$ as functions of $N$ determined entirely by the time dependence for $\epsilon$ that we've obtained from the results of Appendix A. It remains to compute $V$ as a function of $\phi$. To do this, we note that if

$$V(N) = \sum_{n=0} c_n g_n[\phi(N)] \,, \tag{59}$$

where the $g_n$ are some basis of functions,[13] and if $V(N_i)$ and $\phi(N_i)$ are known at $0 \leq i \leq m$ discrete points, then by demanding the expansion for $V(\phi)$ truncate at some order $m$, we obtain a system of $m+1$ equations in $m+1$ unknowns. We can thus solve for the co-efficients $c_i$ for $0 \leq i \leq m$, providing an approximation to the potential to order $m$. For a sufficiently simple $\epsilon$ time dependence, one can go further and explicitly invert (57) to obtain $N$ as a function of $\phi$, substituting back into (58) to obtain $V(\phi)$.

---

[13] e.g. $g_n = \phi^n$ or $g_n = e^{n\kappa\phi}$ for some fixed $\kappa$ etc. The convergence of the procedure detailed above will depend greatly on choice of basis functions adapted.

## C  Effective actions and particle production

In this Appendix, we show how features in the effective potential can be viewed as a sum of terms that include the Coleman-Weinberg correction as the leading adiabatic contribution, plus additional terms corresponding to localised particle production events in the parent theory. What follows closely reproduces the discussion in the appendix of [46]. We begin with the toy example of a heavy field $\psi$ coupled to the inflaton $\phi$:

$$S = \int \sqrt{-g}\Big[\frac{1}{2}\phi\Box\phi - V_{\text{inf}}(\phi)\Big] + \int \sqrt{-g}\Big[\frac{1}{2}\psi\Box\psi - \frac{1}{2}M^2(\phi)\psi^2\Big] + \dots, \tag{60}$$

where we allow for the mass scale $M$ to depend on $\phi$ in a manner that we will specify shortly. We presume that $M$ is parametrically larger than any scale in the action for the inflaton. We integrate out $\psi$ to obtain the (1PI) effective action

$$e^{iW[\phi]} = e^{iS_{\text{inf}}[\phi]} \int \mathcal{D}\psi \, e^{-\frac{i}{2}\int \psi(-\Box+M^2[\phi])\psi} = e^{iS_{\text{inf}}[\phi]}[\det(-\Box+M^2[\phi])]^{-1/2}, \tag{61}$$

where $S_{\text{inf}}[\phi]$ is the first term in (60). Hence

$$W[\phi] = \int \sqrt{-g}\Big[\frac{1}{2}\phi\Box\phi - V_{\text{inf}}(\phi)\Big] + \frac{i}{2}\text{Tr}\log(-\Box+M^2[\phi]). \tag{62}$$

One can evaluate the trace log term in a number of ways. If for instance, $M$ is independent of $\phi$ then the functional determinant can be evaluated exactly (for a heat kernel derivation relevant to the present discussion see the Appendix of [46]), and results in a correction to the potential for $\phi$ of the form

$$V_{\text{eff}}(\phi) = V_{\text{inf}} + V_{\text{ct}} + \frac{M^4}{64\pi^2}\log\Big[M^2/\mu^2\Big]. \tag{63}$$

Here $V_{\text{ct}}$ represents divergent terms that arise given any particular regularisation scheme (e.g. $d-4$ poles in dimensional regularisation)[14] that are to be subtracted by suitable counterterms, and $\mu$ represents the renormalisation scale in a mass independent regularisation scheme (or a cutoff $\Lambda$ would appear in a mass dependent regularisation scheme). The correction term in (63) is the Coleman-Weinberg [82] effective potential. [15] We note that because the functional determinant in (61) was evaluated on a fixed background metric $g_{\mu\nu}$, there are additional curvature corrections to (63) that serve to renormalise the Einstein-Hilbert and cosmological constant terms, in addition to producing higher order curvature corrections that will be suppressed at low energies. In what follows, we presume all couplings to have been fixed by renormalisation conditions.

Recalling (61), we see that we can rewrite (62) as

$$W[\phi] = \int \sqrt{-g}\Big[\frac{1}{2}\phi\Box\phi - V_{\text{inf}}(\phi)\Big] - i\log Z_\psi[\phi], \tag{64}$$

where $Z_\psi$ is given by

$$Z_\psi[\phi] = \int \mathcal{D}\psi \, e^{-\frac{i}{2}\int \psi(-\Box+M^2[\phi])\psi}. \tag{65}$$

---

[14]Since we do not consider derivative interactions between $\phi$ and $\psi$, there is no wave-function renormalisation for $\phi$ up to one loop.

[15]Allowing for $M^2(\phi)$ to depend on $\phi$ will result in derivative corrections to (62) that would reproduce the usual derivative expansion of the effective action.

Hence the (quantum corrected) equations of motion for $\phi$ are obtained from variations of $W[\phi]$, resulting in

$$\Box\phi - V_{\text{inf}}(\phi) = \frac{1}{2}\partial_\phi M^2[\phi]\langle\psi^2\rangle_\phi. \tag{66}$$

Here

$$\langle\psi^2\rangle_\phi \equiv \frac{\int \mathcal{D}\psi \; \psi^2 \; e^{-\frac{i}{2}\int \psi(-\Box+M^2[\phi])\psi}}{\int \mathcal{D}\psi \; e^{-\frac{i}{2}\int \psi(-\Box+M^2[\phi])\psi}}, \tag{67}$$

where evidently the right hand side of (66) is a correlation function of two coincident fields and thus also needs to be suitably regularised. The subscript on the expectation value is to indicate that it is a functional of $\phi$ and its derivatives. If we demand that in the asymptotic past, $M^2[\phi] \to M^2$, where $M$ is a constant heavy scale, and if $(M^2[\phi]-M^2)/M^2 \ll 1$ for all $\phi$, then we can evaluate the above using the Schwinger-Keldysh or in-in formalism in some adiabatic approximation scheme. If the initial state was in the adiabatic vacuum, then the net result of time evolving in the interaction picture would be to leave the state in the adiabatic vacuum if the time evolution of $\phi$ (to be viewed as an external field) is such that $\dot\phi\,\partial_\phi M/M^2 \ll 1$. If this condition is not satisfied, then the vacuum evolves into an excited state described at any given moment by the Bogoliubov coefficients $\alpha_k$ and $\beta_k$ which rotate the mode functions $u_k$ of the $\psi$ field according to:

$$v_k = \alpha_k u_k + \beta_k u_k^*. \tag{68}$$

The hard part of the calculation lies in evaluating these coefficients, but one can still formally proceed assuming this has been done. Since we are dealing with a conservative system, time evolution will not excite modes of arbitrarily high energy, and so the Dyson operator corresponding to (68) evaluated at that moment is given (up to a phase) by the equivalent unitary operator

$$U(\Theta) = e^{-\frac{1}{2}\int[\Theta_k a_k^2 - \Theta_k^* a_k^{\dagger 2}]}. \tag{69}$$

Here $a_k^\dagger$ and $a_k$ are the creation and annihilation operators associated with $\psi$ in the interaction picture, and $\Theta_k \equiv \theta_k e^{i\delta_k}$ are related to the Bogoliubov coefficients (68) as $\alpha_k = \cosh\theta_k$, $\beta_k = e^{-i\delta_k}\sinh\theta_k$. We can therefore formally evaluate $\langle\psi^2\rangle_\phi$ as

$$\langle\psi^2\rangle_\phi = \langle 0|U^\dagger(\Theta)\psi^2 U(\Theta)|0\rangle = \frac{1}{(2\pi)^3}\int \frac{d^3k}{2\omega_k}\Big[1 + 2|\alpha_k\beta_k|\cos(2\delta_k) + 2|\beta_k|^2\Big], \tag{70}$$

where $\omega_k^2 = k^2 + M^2$. The various terms in the square brackets above are familiar to us – the first term contains the Coleman-Weinberg correction. To see this, we bring this contribution to the left hand side of (66), and realising that when $M^2(\phi)$ varies slowly enough, we obtain the correction

$$\frac{\partial_\phi M^2(\phi)}{4(2\pi)^3}\int_{\mu^2}\frac{d^3k}{\sqrt{k^2+M^2}} = \frac{\partial_\phi M^2(\phi)}{32\pi^2}M^2(\phi)\ln\Big[M^2/\mu^2\Big] \equiv \partial_\phi V_{\text{CW}}(\phi). \tag{71}$$

This represents the first term in the adiabatic expansion corresponding to the change of the vacuum energy density of the $\psi$ field along the inflaton trajectory. The second term in the square brackets of (70) is a phase associated with each excited wave number. As discussed in [83, 84] the so-called 'random phase' states (such as thermal states or eigenstates of the number operator) will have contributions that vanish when integrated over. Finally, the last term in (70) is

$$\langle\psi^2\rangle_\phi = \int \frac{d^3k}{\omega_k}\frac{|\beta_k|^2}{(2\pi)^3} \equiv \frac{1}{a^3}\int \frac{d^3k}{\omega_k}\frac{n_k}{(2\pi)^3}, \tag{72}$$

where $n_k$ above is the number density of particles with comoving momenta $k$.

From this, we conclude that the 1PI effective action contains the Coleman-Weinberg correction as the leading adiabatic contribution, with additional contributions that can be interpreted in the parent theory as the transient production of heavy quanta. The real challenge of course is formally calculating the contributions which sum up to (72) since these depend not only on $\phi$, but also its velocity, acceleration, etc and correspond to higher order terms in the adiabatic approximation scheme used in calculating (67). These additional terms will resum to the usual derivative expansion with which we are familiar.

# D  Reconstruction preliminaries

Reconstruction can be viewed in the context of Bayesian inference. Then Tikhonov regularisation, which is a procedure that gives a unique solution to otherwise ill-posed problems [16] and which will be used here, is interpreted as maximum likelihood estimation with a prior on the norm of the squared $n$-th derivative of the quantity to be reconstructed. In this case, the quantity to be reconstructed is the change, possibly fractional, in the EFT parameter function $X(\tau)$ formed into a vector $\mathbf{X}$.

In order to make numerical analysis possible the EFT parameter is first expressed in terms of piecewise constant functions $\phi_j(\tau)$ that are equal to unity between $\tau_j$ and $\tau_{j+1}$ and zero otherwise

$$X(\tau) = \sum_{j=1}^{N} X_j \phi_j(\tau) \,. \tag{73}$$

The components $X_i$ are collected in the vector $\mathbf{X}$. A similar basis $\psi_i(k)$ is constructed for

$$\Delta \mathcal{P}_{\mathcal{R}}/\mathcal{P}_{\mathcal{R}}(k) = \sum_{i=1}^{M} p_i \psi_i(k) \,, \tag{74}$$

and the components $p_i$ are collected in the vector $\mathbf{p}$. Recognising that the relation between $X(\tau)$ and $\Delta \mathcal{P}_{\mathcal{R}}/\mathcal{P}_{\mathcal{R}}(k)$ is linear, a matrix $\mathbf{W_X}$ exists that relates $\mathbf{X}$ to $\mathbf{p}$, namely that

$$\mathbf{p} = \mathbf{W_X} \mathbf{X} \,. \tag{75}$$

The elements $(\mathbf{W_X})_{ij}$ are obtained by calculating $\Delta \mathcal{P}_{\mathcal{R}}/\mathcal{P}_{\mathcal{R}}(k_i)$ due to a 'unit vector' variation $X(\tau) = \phi_j(\tau)$ such that for the speed of sound $1/c_s^2(\tau) - 1 \equiv u(\tau)$, (denoted $\mathbf{u}$ when decomposed) we have

$$(\mathbf{W_u})_{ij} = -k_i \int_{-\infty}^{0} d\tau \, \phi_j(\tau) \sin(2k\tau) = -k_i \int_{\tau_j}^{\tau_{j+1}} d\tau \, \sin(2k_i\tau) \,. \tag{76}$$

Similarly

$$(\mathbf{W_\epsilon})_{ij} = k_i \int_{\tau_j}^{\tau_{j+1}} d\tau \, (k_i\tau)^{-2} [(1 - 2k_i^2\tau^2)\sin(2k_i\tau) - 2k_i\tau\cos(2k_i\tau)] \,, \tag{77}$$

for the fractional change in the slow-roll parameter $\Delta\epsilon/\epsilon(\tau)$ (denoted $\epsilon$ when decomposed).

The uncertainties in the estimated PPS $\hat{\mathbf{p}}$ are described by a covariance matrix $\Sigma_{\mathbf{p}}$. Assuming that this is sufficient to describe the statistics of $\mathbf{p}$, the likelihood is then

$$P(\hat{\mathbf{p}}|\mathbf{p}) = L(\mathbf{p}, \hat{\mathbf{p}}) \propto \exp\left(-(\mathbf{p} - \hat{\mathbf{p}})^T \Sigma_{\mathbf{p}}^{-1} (\mathbf{p} - \hat{\mathbf{p}})/2\right) \,, \tag{78}$$

---

[16]Ill-posed problems either do not have a unique solution or the solution is unstable to small perturbations of the input data.

and inserting (75)

$$P(\hat{\mathbf{p}}|\mathbf{X}) = L(\mathbf{X}, \hat{\mathbf{p}}) \propto \exp\left(-(\mathbf{W_X}\mathbf{X} - \hat{\mathbf{p}})^T \Sigma_{\mathbf{p}}^{-1}(\mathbf{W_X}\mathbf{X} - \hat{\mathbf{p}})/2\right). \tag{79}$$

We use a prior that exponentially suppresses the roughness $R(\mathbf{X})$ of a solution which is given by the integral of the squared derivative of $X$ with respect to $\log \tau$,

$$R\{X(\tau)\} = \int_{-\infty}^{0} d\log\tau \left(\frac{dX}{d\log\tau}\right)^2 = \int_{-\infty}^{0} d\tau \,\tau \left(\frac{dX}{d\tau}\right)^2. \tag{80}$$

Replacing derivatives by finite differences it becomes

$$R(\mathbf{X}) = \sum_{i=1}^{N-1} \tau_i(\tau_{i+1} - \tau_i)\left(\frac{X_{i+1} - X_i}{\tau_{i+1} - \tau_i}\right)^2 = \sum_{i=1}^{N-1} \frac{\tau_i}{\Delta\tau_i}(X_{i+1}^2 - 2X_{i+1}X_i + X_i^2) \equiv \mathbf{X}^T \mathbf{\Gamma}\mathbf{X}, \tag{81}$$

where

$$\mathbf{\Gamma} = \begin{pmatrix} \frac{\tau_1}{\Delta\tau_1} & -\frac{\tau_1}{\Delta\tau_1} & & & \\ -\frac{\tau_1}{\Delta\tau_1} & \frac{\tau_1}{\Delta\tau_1} + \frac{\tau_2}{\Delta\tau_2} & -\frac{\tau_2}{\Delta\tau_2} & & \\ & \ddots & \ddots & \ddots & \\ & & -\frac{\tau_{N-2}}{\Delta\tau_{N-2}} & \frac{\tau_{N-2}}{\Delta\tau_{N-2}} + \frac{\tau_{N-1}}{\Delta\tau_{N-1}} & -\frac{\tau_{N-2}}{\Delta\tau_{N-2}} \\ & & & -\frac{\tau_{N-2}}{\Delta\tau_{N-2}} & \frac{\tau_N}{\Delta\tau_N} \end{pmatrix}. \tag{82}$$

Note that if the penalty on roughness had been on squared derivatives with respect to $\tau$ (linear) then the roughness matrix would have been

$$\mathbf{\Gamma} = \begin{pmatrix} 1 & -1 & & & \\ -1 & 2 & -1 & & \\ & \ddots & \ddots & \ddots & \\ & & -1 & 2 & -1 \\ & & & -1 & 1 \end{pmatrix}. \tag{83}$$

The prior is given by

$$P(\mathbf{X}) \propto \exp(-\lambda R(\mathbf{X})/2) = \exp(-\lambda \mathbf{X}^T \mathbf{\Gamma}\mathbf{X}/2), \tag{84}$$

where $\lambda$ is a (hyper)parameter that controls the prior and is an important parameter in regularisation. The solution will depend on $\lambda$, though since the prior is subjective there is no preferred choice of $\lambda$ except that it should correspond to how much roughness in the EFT parameters one is from the outset willing to accept as plausible. The roughness of the final solution will decrease with $\lambda$ and as $\lambda \to \infty$, $\mathbf{X} \to C$ where $C$ is a constant.

With the likelihood and the prior given the posterior is

$$P(\mathbf{X}|\hat{\mathbf{p}}) \propto P(\hat{\mathbf{p}}|\mathbf{X})P(\mathbf{X}) = \exp(-(\mathbf{W_X}\mathbf{X} - \hat{\mathbf{p}})^T \Sigma_{\mathbf{p}}^{-1}(\mathbf{W_X}\mathbf{X} - \hat{\mathbf{p}})/2 - \lambda \mathbf{X}^T \mathbf{\Gamma}\mathbf{X}/2), \tag{85}$$

and its maximum is also the minimum of

$$Q(\lambda) = -2\log P(\mathbf{X}|\hat{\mathbf{p}}) = (\mathbf{W_X}\mathbf{X} - \hat{\mathbf{p}})^T \Sigma_{\mathbf{p}}^{-1}(\mathbf{W_X}\mathbf{X} - \hat{\mathbf{p}})/2 + \lambda \mathbf{X}^T \mathbf{\Gamma}\mathbf{X}/2, \tag{86}$$

which in this Gaussian case is

$$\hat{\mathbf{X}} = (\mathbf{W_X}^T \Sigma_{\mathbf{p}}^{-1}\mathbf{W_X} + \lambda\mathbf{\Gamma})^{-1}\mathbf{W_X}^T \Sigma_{\mathbf{p}}^{-1}\hat{\mathbf{p}} \equiv \mathbf{M}\hat{\mathbf{p}}. \tag{87}$$

This is a linear map $\mathbf{M}$ from the data $\hat{\mathbf{p}}$ to the solution $\hat{\mathbf{X}}$. The inverse Hessian $\mathbf{H}^{-1}$ which gives the Bayesian covariance matrix $\Sigma_{\mathrm{F}}$, or credible intervals of $\mathbf{X}$, is

$$\mathbf{H}^{-1} = (\mathbf{W}_{\mathbf{X}}^T \Sigma_{\mathbf{p}}^{-1} \mathbf{W}_{\mathbf{X}} + \lambda \Gamma)^{-1} . \tag{88}$$

In the frequentist picture, the starting point is also (86) but it is now seen as a penalised log likelihood with a penalty $\lambda \mathbf{X}^T \Gamma \mathbf{X}$ on roughness. The maximum likelihood solution is the same as before, namely (87), but the error analysis is different. In the frequentist view, there is nothing special about the data $\hat{\mathbf{p}}$ and so other realisations of the data should be considered. The error propagation from $\mathbf{p}$ to $\mathbf{X}$ must then be determined. Since there is a linear map represented by the matrix $\mathbf{M}$ between $\mathbf{p}$ and $\mathbf{X}$ the covariance matrix of $\mathbf{p}$, $\Sigma_{\mathbf{p}}$ is related to $\Sigma_{\mathrm{F}}$, the frequentist covariance matrix of $\mathbf{X}$, by a similarity transformation

$$\Sigma_{\mathrm{F}} = \mathbf{M} \Sigma_{\mathbf{p}} \mathbf{M}^T. \tag{89}$$

In performing the reconstructions a choice of $\lambda$ must necessarily be made. As $\lambda$ is increased the reconstructions will be biased, *i.e.*, there is a systematic shift of the mean of the reconstructions away from the true solution, but the variance of the reconstructions will decrease. As $\lambda$ is decreased, the variance increases but the bias decreases.

The roughness is given by

$$\int_{-\infty}^{0} d \log \tau \left( \frac{d \epsilon(\tau)}{d \log \tau} \right)^2 \to \epsilon^T \Gamma \epsilon, \tag{90}$$

where $\Gamma$ is the roughness matrix, the discretised form of the differential operator in the integral (90). Equivalently, the reconstruction is the minimum of (twice) the penalised negative log likelihood

$$Q(\lambda) = (\mathbf{W}_\epsilon \epsilon - \hat{\mathbf{p}})^T \Sigma_{\mathbf{p}}^{-1} (\mathbf{W}_\epsilon \epsilon - \hat{\mathbf{p}}) + \lambda \epsilon^T \Gamma \epsilon, \tag{91}$$

where $\hat{\mathbf{p}}$ is the estimated fractional PPS change, $\Sigma_{\mathbf{p}}$ is the covariance matrix of the fractional PPS change and $\lambda$ is the regularisation parameter which controls the degree to which a solution with roughness is disadvantaged with a high penalty, and consequent preference for smooth solutions when a high value of $\lambda$ is chosen.

For a given $\lambda$ the minimum of (95) is

$$\hat{\epsilon} = (\mathbf{W}_\epsilon^T \Sigma_{\mathbf{p}}^{-1} \mathbf{W}_\epsilon + \lambda \Gamma)^{-1} \mathbf{W}_\epsilon^T \Sigma_{\mathbf{p}}^{-1} \hat{\mathbf{p}} \equiv \mathbf{M} \hat{\mathbf{p}}, \tag{92}$$

where $\mathbf{M}$ maps between $\hat{\mathbf{p}}$ and $\hat{\epsilon}$. The speed of sound case is discussed in Appendix D.1.

Given that this reconstruction method has a statistical interpretation the uncertainties of the solution are clearly defined. The Bayesian covariance matrix is the inverse Hessian

$$\Sigma_{\mathrm{B}} = (\mathbf{W}_\epsilon^T \Sigma_{\mathbf{p}}^{-1} \mathbf{W}_\epsilon + \lambda \Gamma)^{-1}, \tag{93}$$

and the frequentist covariance matrix

$$\Sigma_{\mathrm{F}} = \mathbf{M} \Sigma_{\mathbf{p}} \mathbf{M}^T, \tag{94}$$

originates from the propagation of uncertainties from the data $\hat{\mathbf{p}}$ described by covariance matrix $\Sigma_{\mathbf{p}}$ to the solution $\hat{\epsilon} = \mathbf{M} \hat{\mathbf{p}}$, which, since the relation is linear, is given by a similarity transformation.

The reconstruction of $\mathbf{u}$ has the additional complication that $\mathbf{u}$ should be everywhere non-negative which is a constraint that makes the likelihood non-Gaussian and a numerical solution necessary. This case is discussed in Appendix D.1.

### D.1  Positivity of EFT parameter changes

The posterior probability distribution (85) is Gaussian in $\mathbf{X}$ and so the maximum likelihood solution $\hat{\mathbf{X}}$ may take negative values depending on the data $\hat{\mathbf{p}}$. However, the speed of sound $c_s$ should not be greater than unity, or rather the departure of the inverse square speed of sound from unity $u(\tau) = 1/c_s^2(\tau) - 1$ must be positive. To impose this constraint, we minimise instead twice the negative log likelihood

$$Q(\lambda) = -2\log P(\mathbf{v}|\hat{\mathbf{p}}) = (\mathbf{W_u}\exp(\mathbf{v}) - \hat{\mathbf{p}})^T \Sigma_{\mathbf{p}}^{-1} (\mathbf{W_u}\exp(\mathbf{v}) - \hat{\mathbf{p}}) + \lambda \exp(\mathbf{v})^T \Gamma \exp(\mathbf{v}), \quad (95)$$

with respect to the vector $\mathbf{v}$ where $\mathbf{u} = \exp(\mathbf{v})$. Then even if a $\mathbf{v}$ with negative entries was found as the minimum $\exp(\mathbf{v})$ will be positive. As this likelihood is non-Gaussian with respect to $\mathbf{v}$ no analytic solution of $\hat{\mathbf{v}}$ given $\hat{\mathbf{p}}$ exists. Instead, a solution can be obtained by numerical minimisation using gradient quasi-Newton methods such as BFGS. The uncertainties on $\hat{\mathbf{u}} = \exp(\hat{\mathbf{v}})$ are evaluated as in the previous case.

### D.2  Sidestepping the PPS

The EFT parameter $\mathbf{X}$ is estimated from the PPS $\hat{\mathbf{p}}$ which is *itself* an estimate from data. Though not done here, it is possible to reconstruct the EFT parameter $\mathbf{X}$ *directly* from a data set $\mathbf{d}$. Given a linear relation between the PPS $\mathbf{p}$ and the data set $\mathbf{d}$ such that

$$\mathbf{d} = \mathbf{W}\mathbf{p}, \quad (96)$$

and given that

$$\mathbf{p} = \mathbf{W_X}\mathbf{X}, \quad (97)$$

then

$$\mathbf{d} = \mathbf{W}(\mathbf{W_X}\mathbf{X}) \equiv \mathbf{W}'\mathbf{X}. \quad (98)$$

Here the new transfer function,

$$\mathbf{W}' = \mathbf{W}\mathbf{W_X}, \quad (99)$$

can now be used instead of $\mathbf{W}$ in the procedure of reconstructing $\mathbf{p}$ from $\mathbf{d}$ with the only difference that now $\mathbf{X}$ will be reconstructed from $\mathbf{d}$ using $\mathbf{W}'$. This procedure is more correct as it collapses two regularisations, obtaining $\hat{\mathbf{p}}$ from $\mathbf{d}$ and then obtaining $\hat{\mathbf{X}}$ from $\hat{\mathbf{p}}$, into one. If done in two steps there is a bias due to the penalty term introduced in each step which is not easily quantified and there are furthermore two regularisation parameters $\lambda_{\mathbf{p}}$ and $\lambda_{\mathbf{X}}$ to take into account. With the collapse and the use of $\mathbf{W}'$ to reconstruct $\mathbf{X}$ from $\mathbf{d}$ there is only one regularisation parameter and all preceding formulae can be used to correctly account for the uncertainties in $\mathbf{X}$ as they are induced by the uncertainties in $\mathbf{d}$.

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
