# Peer review of "Reconstructing the EFT of Inflation from Cosmological Data"

_SciPost Physics, doi:SciPost Phys. 7, 049 (2019)_

## Round 1 · Referee Report · Anonymous (Referee 1) · 2019-5-29

Strengths

  1. Well-written and explained
  2. Valuable result allowing accurate (percent level) reconstruction of potential features in the primordial power spectrum from (single-clock) EFT models of inflation
  3. Good and clear analytic toy-model examples providing proof of principle

Weaknesses

  1. Discussion of the phase transitions and in particular the additional sharp oscillations that need to be introduced for the Einstein-de Sitter background results.
  2. Explanation and discussion of the leading order corrections to the potential

Report

I enjoyed reading this paper, which is well-written and very clear. The authors show convincingly that they can reconstruct, to second order, estimated features in the primordial power spectrum from time dependent parameters in and EFT description of single-clock models of inflation. First establishing the relations to second order in the features, they apply their method to an analytic toy model example, providing a proof of principle. They then apply their method to a Lambda-CDM and Einstein-de Sitter background model, resulting in 2nd-order reconstructed slow-roll epsilon parameters and the inflationary potential. Their method is a valuable contribution that will allow a more precise reconstruction of (and further constrain the presence of) features, assuming a particular background model. I would have appreciated a more elaborate discussion on the meaning and implications of the reconstruction in the Einstein-de Sitter case, where the result for the slow-roll parameter has more fine-structure and in particular seems to require a sharp oscillation over less than a single e-fold. To explain better how the (claimed) significance of this sharp oscillation compares to the (larger) bumps would be useful. In the discussion, which is very brief, the authors introduce an example of leading order adiabatic corrections to the effective potential that might give rise to the reconstructed features. This paragraph would benefit from a bit more context (instead of just referring to the appendix) and a more specific discussion with respect to the two background scenario's discussed (\Lambda CDM versus Einstein-de Sitter).

Besides these relatively minor points, which I would like to see addressed, I would recommend the manuscript to be published.

Requested changes

  1. Address the significance of the sharp oscillation in the Einstein-de Sitter scenario
  2. Expand the discussion on the leading corrections to the effective potential and in particular what that means for the \Lambda CDM and Einstein-de Sitter background results

  • validity: high
  • significance: good
  • originality: ok
  • clarity: high
  • formatting: good
  • grammar: excellent

Author:  Subodh Patil  on 2019-07-16  [id 563]

(in reply to Report 1 on 2019-05-29)
Category:
answer to question

We are grateful to the Referee for the encouraging and helpful comments.

We have noted the concern regarding the significance of the fine structure in our reconstructed slow roll parameters and accordingly reexamined our procedure for estimating the error bands. This turned out to be an important exercise because we realised that we had underestimated them by considering only the diagonal elements of the frequentist covariance matrix which describes the error in the reconstructed PPS. We have now repeated our calculations using the the full covariance matrix and find a much enlarged error band. (All this took some time hence the delay in responding - however this is not yet a complete analysis which would require evaluating the full likelihood … which is beyond the scope of the present work.) We have attached a note which shows graphically the significant correlations at both high and low wave numbers which result in the enlarged uncertainties.

This of course does not undermine our main result which is to demonstrate accurate EFT parameter reconstruction from a cosmological data set. It does however enable us to (hopefully) adequately address the Referee’s remark that “… in the Einstein-de Sitter case … the result for the slow-roll parameter has more fine-structure and in particular seems to require a sharp oscillation over less than a single e-fold. To explain better how the (claimed) significance of this sharp oscillation compares to the (larger) bumps would be useful”. We agree and have now added explanatory text to both “1. Address the significance of the sharp oscillation in the Einstein-de Sitter scenario” as well as to "2. Expand the discussion on the leading corrections to the effective potential and in particular what that means for the \Lambda CDM and Einstein-de Sitter background results” as the Referee has requested.

For ease of identifying the added sections we have marked these in blue in the attached manuscript (including some additional references). We hope the Referee is satisfied with these clarifications and our revised manuscript can now be published in SciRep.

Attachment:

note_changes_highlighted.pdf

---

## Round 2 · Author Response

List of changes
We have added to the discussion of the results section for Lambda-CDM, as well as the Einstein- de Sitter case in addition to expanding the discussion section which elaborates on how the required small corrections to the potential that would reproduce the requisite features arise as an effective action. We have also added some references.

---

## Round 2 · List of Changes

We have added to the discussion of the results section for Lambda-CDM, as well as the Einstein- de Sitter case in addition to expanding the discussion section which elaborates on how the required small corrections to the potential that would reproduce the requisite features arise as an effective action. We have also added some references.

---

## Editorial Decision

published